# Comparison of Two Photolytic Calibration Methods for Nitrous Acid

Andrew J. Lindsay and Ezra C. Wood

Department of Chemistry, Drexel University, Philadelphia, PA, USA

*Correspondence to*: Ezra C. Wood (ew456@drexel.edu)

**Abstract.** Nitrous acid (HONO) plays an important role in tropospheric oxidation chemistry as it is a precursor to the hydroxyl radical (OH). Measurements of HONO have been difficult historically due to instrument interferences and difficulties in sampling and calibration. The traditional calibration method involves generation of HONO by reacting hydrogen chloride vapor with sodium nitrite followed by quantification by various methods (e.g., conversion of HONO to nitric oxide (NO) followed by chemiluminescence detection). Alternatively, HONO can be generated photolytically in the gas-phase by reacting NO with OH radicals generated by $H_2O$ photolysis. In this work, we describe and compare two photolytic HONO calibration methods that were used to calibrate an iodide adduct chemical ionization mass spectrometer (CIMS). Both methods are based on the water vapor photolysis method commonly used for OH and $HO_2$ (known collectively as HOx) calibrations. The first method is an adaptation of the common chemical actinometry HOx calibration method, in which HONO is calculated based on quantified values for $[O_3]$, $[H_2O]$, $[O_2]$, and the absorption cross sections for $H_2O$ and $O_2$ at 184.9 nm. In the second, novel method HONO is prepared in mostly $N_2$ ($[O_2] = 0.040$ %) and is simply quantified by measuring the $NO_2$ formed by the reaction of NO with $HO_2$ generated by $H_2O$ photolysis. Both calibration methods were used to prepare a wide range HONO mixing ratios between of ~400 and 8,000 pptv. The uncertainty of the chemical actinometric calibration is 27 % ($2\sigma$) and independent of HONO concentration. The uncertainty of the $NO_2$ proxy calibration is concentration-dependent, limited by the uncertainty of the $NO_2$ measurements. The $NO_2$ proxy calibration uncertainties ($2\sigma$) presented here range from 4.5 to 24.4 % (at [HONO] = 8,000 pptv and [HONO] = 630 pptv, respectively) with a 10 % uncertainty associated with a mixing ratio of ~1,600 pptv, typical of values observed in urban areas at night. We also describe the potential application of the $NO_2$ proxy method to calibrating HOx instruments (e.g., LIF, CIMS) at uncertainties below 15 % ($2\sigma$).

## 1 Introduction

Nitrous acid (HONO) is a source of the most important atmospheric oxidant – the hydroxyl radical (OH) – and can therefore play an important role in tropospheric oxidation chemistry. The hydroxyl radical initiates the removal of most trace gases from the atmosphere leading to the formation of secondary pollutants such as ozone ($O_3$) and secondary aerosols. Photolysis of HONO yields OH and nitric oxide (NO):

HONO + hv → OH + NO  ($\lambda < 400$ nm) (R1)

This reaction is the primary sink of HONO during the daytime leading to a typical chemical lifetime at mid-day of between 10-20 minutes at mid-latitudes. Sources of HONO include homogeneous formation (R2), direct emissions from combustion (vehicles, biomass burning, etc.) and soils, and numerous heterogenous processes including heterogeneous reaction of $NO_2$ with moist terrestrial surfaces, photolysis of particulate nitrate (Ye et al., 2016; Ye et al., 2017), and photolysis of nitric acid (Ye et al., 2016).

OH + NO + M → HONO + M (R2)

The relative importance of these sources varies with environment (Jiang et al., 2022).

HONO photolysis has been reported as a major source of HOx (HOx = OH + HO$_2$) throughout the day in a variety of environments, including urban and highly polluted areas (Whalley et al., 2018; Slater et al., 2020; Ren et al., 2013; Lu et al., 2019) as well as more pristine environments (Villena et al., 2011; Jiang et al., 2020; Bloss et al., 2007). Vertical distributions of HONO, however, indicate that its significance as a HOx precursor may be limited to near ground level (Li et al., 2014; Young et al., 2012; Villena et al., 2011; Wong et al., 2012; Tuite et al., 2021; Jaeglé et al., 2018). HONO can also serve as an important source of HOx in indoor environments since sufficient UV light can penetrate windows and substantial HONO concentrations can result from various activities (e.g., cooking) (Gomez Alvarez et al., 2013; Wang et al., 2020).

HONO is notoriously difficult to measure. It can be formed via heterogeneous chemistry within sampling lines or an instrument's inlet. The resulting interferences may pose additional challenges in applying an instrument's zero or in calibration processes that alter the interfering species. Some intercomparison studies have shown substantial differences between HONO measurement techniques. A comparison of several HONO measurements in Beijing showed an overall mixed agreement with a few instruments disagreeing by more than a factor of two (Crilley et al., 2019). Measurements in Houston, Texas showed overall good agreement (within 20 %) between most instruments with larger differences of over 100 % observed for one of the instruments for some time periods (Pinto et al., 2014). Bourgeois et al. (in review, 2022) recently reported an 80 % difference between HONO measurements made by cavity-enhanced spectroscopy and iodide-adduct chemical ionization mass spectrometry (CIMS). Closer agreement for two instrument HONO comparisons has been reported by Stutz et al. (2010) (comparing differential optical absorption spectroscopy (DOAS) and mist-chamber ion chromatography (IC)), Cheng et al. (2013) (comparing Long-path absorption photometry (LOPAP) and stripping coil IC), and Dixneuf et al. (2022) (comparing LOPAP and cavity-enhanced absorption spectroscopy), though many of these studies report considerable deviations when HONO mixing ratios were less than ~100 pptv.

Calibrations for HONO are challenging as this compound is not commercially available and rather must be prepared in situ. Most commonly, HONO is prepared by reacting hydrogen chloride vapor with sodium nitrite (Febo et al., 1995):

$$HCl_{(g)} + NaNO_{2(s)} \rightarrow HONO_{(g)} + NaCl_{(s)} \tag{R3}$$

This method presents several challenges. A stable source of HCl is required, usually from a heated aqueous solution, a gas cylinder, or a permeation tube. Consistent mixing between the HCl and the NaNO$_2$ powder is required. These calibrations also require substantial warmup times (often hours) to ensure source stability, though some recent versions report faster warmup periods (e.g., < 10 min reported by Villena and Kleffmann (2022)). High HONO concentrations (above 1 ppmv) are often produced, requiring dilution, though the temporary unrealistic HONO concentrations can lead to significant HONO loss by its self-reaction and inaccurate HONO quantification. A recent, noteworthy version of this calibration improves upon this concentration issue and has the ability to produce [HONO] on the order of tens of pptv (Lao et al., 2020). The generated HONO can be quantified by various methods including theoretical calculation (Villena and Kleffmann, 2022), conversion to NO followed by chemiluminescence detection (Lee et al., 2012; Lao et al., 2020; Villena and Kleffmann, 2022), thermal conversion to NO$_2$ followed by NO$_2$ quantification (Gingerysty and Osthoff, 2020), and conversion to aqueous nitrite followed by derivatization and detection by UV-vis (Peng et al., 2020). The calibration uncertainty depends on the output stability of the HONO source and the quantification technique used. Villena and Kleffmann (2022) demonstrate using two separate techniques that overall calibration uncertainties can be well below 10 % (2σ).

More recently, photolytic HONO sources have been utilized. Humidified air is exposed to ultraviolet (UV) light to photolyze $H_2O$ to produce an equal mixture of OH and $HO_2$, which in the presence of excess NO then converts to HONO. This HONO output is stable within seconds (i.e., the initial UV lamp warm up time) and is tunable by altering humidity, UV flux, or UV exposure time. The HONO formed has been quantified based on the water vapor mixing ratio, water vapor absorption cross section, the UV flux, and the UV exposure time. This quantification approach thus far has been used to calibrate HONO photo-fragmentation instruments that detect OH using laser-induced fluorescence spectroscopy (Dyson et al., 2021; Bottorff et al., 2021). The HONO formed from a photolytic source has also been quantified by thermal dissociation followed by measurement of the $NO_2$ produced (Veres et al., 2015). These methods have an uncertainty of 30 to 36% ($2\sigma$), similar to the uncertainty for HOx calibrations based on water vapor photolysis (Dusanter et al., 2008). In this manuscript, we present an alternative photolytic HONO calibration that we refer to as the "$NO_2$ proxy" method. This method requires a direct $NO_2$ measurement that is used as a 'proxy' to quantify HONO concentrations. We compare this new proxy calibration to the more standard photolytic calibration method as performed by Bottorff et al. (2021) and Dyson et al. (2021). This method has a lower uncertainty (typically ~10%, $2\sigma$) and unlike the actinometric method does not require characterization of the mercury lamp emission spectrum.

## 2 Methods

### 2.1 Instrumentation

A Cavity Attenuated Phase Shift (CAPS) spectrometer (Aerodyne Research, Inc.) was used to detect $NO_2$ (Kebabian et al., 2008). The CAPS also indirectly measured $O_3$ as it was converted to $NO_2$ by reaction with excess NO. The CAPS instrument was calibrated using a 2B Technologies Model 306 $O_3$ Calibration Source. Ozone outputs were varied between 10 and 300 ppbv with greater than 99.99 % conversion efficiency to $NO_2$ by reaction with excess NO ([NO] = 1.82 ppmv) within approximately 15 m of FEP tubing (i.d. = 0.476 cm; residence time = 17.1 s, pseudo-first order rate constant of 0.8 $s^{-1}$). The manufacturer stated accuracy of this $O_3$ calibrator is 2 % ($2\sigma$), though no recent factory calibrations have been conducted. Therefore, a second calibration was conducted with a Thermo Environmental Instruments 49C $O_3$ Calibrator, which agreed to within 2.5 %. We assign an uncertainty of 4 % ($2\sigma$) to the $NO_2$ measurements to account for possible drift in accuracy. Temperature and relative humidity (RH) measurements were made using two Vaisala HMP60 probes and used, along with pressure measurements, to calculate $H_2O_{(g)}$ mixing ratios. Both probes agreed with a new RH/T replacement sensor (manufacturer stated uncertainty: 3 %) to within 3 %. We assign an uncertainty of 5 % ($2\sigma$) to our determined $H_2O$ mixing ratios.

A High-Resolution Chemical Ionization Time of Flight Mass Spectrometer (HR-ToF-CIMS, Tofwerks/Aerodyne Research, Inc.) was used to detect HONO (Bertram et al., 2011; Lee et al., 2014). HONO and concomitant gases are ionized within a laboratory-built ion-molecule reactor (IMR) using reagent iodide ($I^-$) ions. Our lab-built IMR is internally coated with PTFE and sampled the calibration gas at a flow rate of 2.10 SLPM through a stainless-steel critical orifice (nominal diameter of 0.48 mm). Iodide ($I^-$) reagent ions in $N_2$ (Airgas, industrial grade) were sampled at 2.20 SLPM through a similar critical orifice perpendicular to the main sample flow. The $I^-$ was prepared by exposing dilute methyl iodide ($CH_3I$) from a permeation tube (VICI Metronics) to a $^{210}Po$ radioactive source. Humidified $N_2$ was also added to the IMR perpendicular to the main sample flow at a flow rate of 0.365 SLPM. The pressure in the IMR was held at 80 mbar, controlled by adjusting a valve to a scroll pump (Agilent Technologies IDP-7).

Ions are separated by mass-to-charge ratio (m/z) at a mass resolving power of near 5000 m/$\Delta$m. We monitor the HONO iodide adduct $I(HONO)^-$ at 173.90575 m/z. The peak-fitting software (TofWare) accounts for the overlapping contribution from the $^{13}C$

formic acid $I(^{13}CH_2O_2)^-$ peak at 173.91342 m/z. We account for the humidity dependence of the instrumental response by determining the mole fraction of $H_2O_{(g)}$ ($\chi_{H2O}$) in the IMR by measuring the RH and temperature of the IMR in the exhaust of the scroll pump. See the supplement for more information regarding humidity effects for the HONO ionization chemistry (see Sect. S1.2). Analytical parameters including the limit of detection, precision, and linear range of these HONO measurements are also detailed within the supplement (see Sect. S1.1).

## 2.2 Calibration Methods

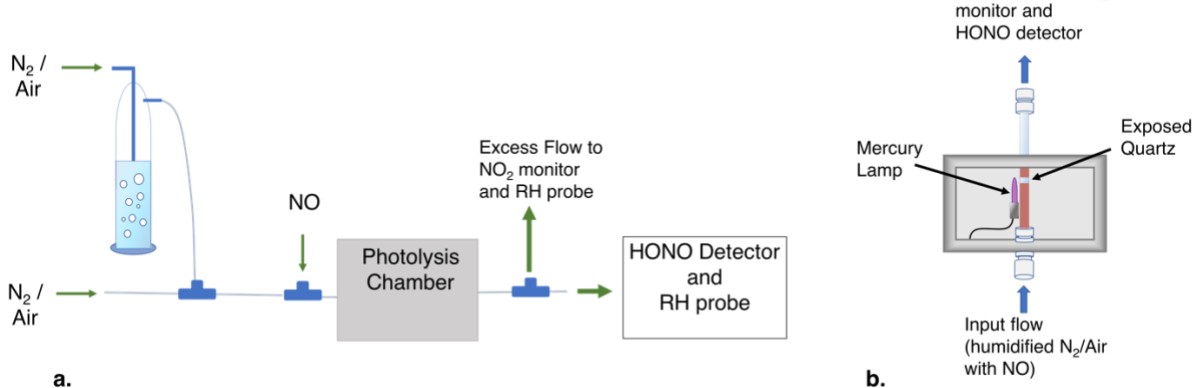

**a.**                                                                                                    **b.**

Figure 1: Schematic of the experimental setup (a.) and the photolysis chamber (b.). Not shown: small flow of air when using $N_2$ as the carrier gas (a.), and the purge flow of $N_2$ (b.).

We calibrate HONO using two variations of the water vapor HOx calibration method: one is a modification of the standard actinometric HOx photolytic calibration and the other we refer to as the "$NO_2$ proxy" calibration. These calibration methods mainly differ in how HONO is quantified. In both methods, HONO was produced nearly identically. Air (Airgas, Ultra Zero grade; $[O_2]$ = 21 ± 1 %) for the actinometric method or $N_2$ (Airgas, industrial grade) for the $NO_2$ proxy calibration is humidified with HPLC (high-performance liquid chromatography) grade water (Fisher Chemical), mixed with NO (Airgas, 41.02 ± 2.05 ppmv in $N_2$), and then exposed to 184.9 nm ultraviolet radiation from a low-pressure mercury lamp (Jelight 78-2046-1). While in the experiments presented in this manuscript we used industrial grade $N_2$ for the humidified CIMS IMR inflow (mentioned in Sect. 2.1) and as the $NO_2$ proxy calibration carrier gas, we have used ultra-high purity $N_2$ (Airgas) in previous experiments. We find no differences between the calibration results acquired using different grades of $N_2$. The resulting OH and $HO_2$ from water photolysis form HONO by reaction with excess NO (R4-6a and R2).

$$H_2O + h\nu \rightarrow H + OH \tag{R4}$$

$$H + O_2 + M \rightarrow HO_2 + M \tag{R5}$$

$$HO_2 + NO \rightarrow OH + NO_2 \tag{R6a}$$

$$OH + NO + M \rightarrow HONO + M \tag{R2}$$

The $HO_2$ to HONO pathway is limited by the small fraction of R5 that forms $HNO_3$ rather than OH and $NO_2$:

$$HO_2 + NO + M \rightarrow HNO_3 + M \tag{R6b}$$

A schematic of the setup used for both calibrations is shown in Fig. 1. The mercury lamp is housed within a 10.8 cm × 26.7 cm × 10.2 cm photolysis chamber (Fig. 1b), and the volume surrounding the lamp is purged with dry $N_2$ (purge not shown). The

humidified air-NO mixture is transported past the mercury lamp within a partially exposed quartz tube (I.D. = 1.04 cm, total length = 26.7 cm; exposed length = ~ 0.5 cmp). HONO sample concentrations are controlled by adjusting the lamp flux with a Variac Variable transformer, adjusting the relative flow rates of the dry and humidified zero air/$N_2$, or adjusting the absolute flow rates to alter the lamp exposure time. For our example calibrations discussed in this manuscript, we typically used a main $N_2$ or air flow rate of 5 SLPM with an addition of 200 sccm of 41.02 ppmv NO in $N_2$ for a total flow rate of 5.20 SLPM and a diluted NO mixing

ratio of 1.58 ppmv. Pseudo-first order rate constants calculated using this [NO] for R6a and R2 are 322 s$^{-1}$ and 295 s$^{-1}$, respectively. Under these conditions, HOx is converted to HONO within 0.02 s inside the remaining 11.4 cm of the quartz tube. The [NO] chosen must be high enough to minimize OH and $HO_2$ wall losses. We have ensured that this NO mixing ratio is sufficient in separate experiments by confirming that no additional HONO signal results at increased [NO] values. Possible HONO formation by additional photolytic processes (specifically involving the surfaces of the quartz photolysis tube) was tested by monitoring

CIMS I(HONO)$^-$ signals during additional experiments. These experiments include exposing dry carrier gas ([NO] = 1.58 ppmv) to 184.9 nm radiation (i.e., exposure to UV without $H_2O$ photolysis), exposing humidified carrier gas ([NO] = 0 ppmv) to 184.9 nm radiation (i.e., to investigate if HONO is formed by heterogenous reactions involving $H_2O$ or HOx with NO, $NO_2$, or $HNO_3$ adsorbed on the quartz tube), and exposing humidified carrier gas ([NO] = 1.58 ppmv) to the 254 nm radiation from a separate mercury lamp (Jelight 81-3306-2) in which the 184.9 nm emission is blocked (i.e., the carrier gas matches calibration conditions

and is exposed to UV radiation but without $H_2O$ photolysis). These tests indicate no appreciable HONO formation by other photolytic processes.

The resulting calibration gas enters a PFA tee and is arranged so that the air travels straight to the CIMS (2.1 SLPM) while the remaining flow (~ 3.1 SLPM) makes a 90º turn for the CAPS line which includes a vent. The gas flow is initially laminar within the quartz photolysis tube (Reynolds number ≈ 600). This results in an initial [$HO_2$] (and therefore [HONO]) radial gradient in

which the greatest concentrations exist near the flow tube walls (i.e., where the flow rates are lower and the UV exposure times longer). Turbulence is induced by the sudden changes in tube inner diameter at the quartz tube exit (reducing union) and upon entering the PFA tee. The air is therefore most likely well mixed prior to being split within the PFA tee. The excess flow within the CAPS line (~ 2 SLPM) was vented past an RH/T probe to determine the water mixing ratio in the photolysis cell. A second RH/T probe quantified the water mixing ratio in the CIMS IMR as previously mentioned in Sect. 2.1. Details of the two calibration

methods are described in the following sections.

### 2.2.1 Actinometric Calibration

The water vapor photolysis calibration method has been used for several decades to calibrate OH and $HO_2$ measurements (Stevens et al., 1994; Lanzendorf et al., 1997; Dusanter et al., 2008). The concentration of HOx, and therefore HONO, is calculated from the time-integrated photolysis of water vapor:

$$[HONO] \approx [HOx] = (F \cdot t)[H_2O]\sigma_{H_2O}\phi_{HO_x} \tag{1}$$

where $F$ is the photon flux at 184.9 nm, $t$ is the UV irradiation time, $\sigma_{H2O}$ is the absorption cross section of water at 184.9 nm, and $\Phi_{HOx}$ is the quantum yield of HOx from water photolysis and equal to 2. $F$ can be quantified using direct actinometric measurements (e.g., using a calibrated phototube), and $t$ can be quantified via characterization of the flow rates and photolysis cell geometry (Faloona et al., 2004). Alternatively, and more commonly among HOx measurement groups, the product $F \cdot t$ can be determined via

"chemical actinometry" (Schultz et al., 1995). In the $O_2$-$O_3$ chemical actinometry method, the concentration of $O_3$ produced by photodissociation of $O_2$ at 184.9 is used to determine $F \cdot t$:

$$O_2 + hv \rightarrow O + O \tag{R7}$$

$$O + O_2 + M \rightarrow O_3 + M \tag{R8}$$

The product of the lamp flux and the exposure time, i.e., the $(F{\cdot}t)$ term, is given by Eq. (2), in which $\sigma_{O_2}$ is the absorption cross section of $O_2$ at 184.9 nm and $\Phi_{O_3}$ is quantum yield of $O_3$ from $O_2$ photolysis ($\Phi_{O3} = 2$):

$$(F \cdot t) = \frac{[O_3]}{[O_2]\sigma_{O_2}\phi_{O_3}} \tag{2}$$

Substituting this expression for $F{\cdot}t$ into Eq. (1) gives Eq. (3):

$$[HONO] \approx [HOx] = \frac{[O_3]}{[O_2]\sigma_{O_2}}[H_2O]\sigma_{H2O} \tag{3}$$

The effective value for $\sigma_{O2}$ must be experimentally determined for the individual mercury lamp at the experimental $O_2$ optical depth. This is required because the emission profile near 14.9 nm, which comprises two peaks due to self-reversal, can vary from lamp to lamp and with operating conditions, and the $O_2$ absorption spectrum steeply decreases near the mercury lamp emission maximum (Lanzendorf et al., 1997). We use an experimentally determined $\sigma_{O2}$ value of $1.4 \times 10^{-20}$ cm$^2$ molec$^{-1}$ for the mercury lamp used for these experiments. The JPL-recommended value of $7.1 \times 10^{-20}$ cm$^2$ molec$^{-1}$ was used for $\sigma_{H2O}$ (Burkholder et al., 2020). The value of $[O_2]$ is based on the flows mentioned in Sect. 2.2 and is equal to $20.1 \pm 1.0$ %. The $O_2$ optical depth is 0.033, and the $O_2$ column density (within the photolysis tube center) is $2.4 \times 10^{18}$ molecules cm$^{-2}$. For typical operating conditions, the value of $F{\cdot}t$ and an estimated photon flux $F$ (calculated using an approximate gas exposure time) are $3.48 \times 10^{12}$ photons cm$^{-2}$ and $7.1 \times 10^{14}$ photons cm$^{-2}$ s$^{-1}$, respectively. The value of $[O_3]$ here was near 20 ppbv and determined with the CAPS $NO_2$ monitor after its reaction with NO, forming $NO_2$. This is measured with dry air flowing in the photolysis chamber so that the $NO_2$ measured is solely from the reaction of NO with $O_3$ and not $HO_2$. These $F{\cdot}t$ and $[O_3]$ values are high compared to those used for most $O_3$ actinometry HOx calibrations, in which $[O_3]$ is often less than 1 ppbv (e.g., Faloona et al. (2004)), but are comparable to those used by Dusanter et al. (2008). High $F{\cdot}t$ values were used so that typical ambient HONO concentrations (ranging up to several ppbv) could be prepared. $[H_2O]$ is determined using the measured RH, temperature, and pressure. The uncertainties of the variables in Eq. (3) are discussed in Sect. S3 of the supplement. The combined uncertainty ($2\sigma$) for [HOx] (and therefore [HONO]) calculated using this equation is 27 % (see supplement for details).

We apply a small correction to the value for [HOx] calculated in Eq. (3) in order to obtain [HONO]. This correction accounts for the incomplete conversion of $HO_2$ to HONO due to R6b.

$$[HONO] = (0.5 + 0.5 \cdot \frac{1}{1+\beta}) \cdot [HOx] \tag{4}$$

Equation 4 includes the variable $\beta$, which is the relative rate or product ratio of R6b to R6a (i.e., $\beta = k_{R6b}/k_{R6a} = [HNO_3]/[NO_2]$) and depends on temperature, pressure, and humidity (Butkovskaya et al., 2007; Butkovskaya et al., 2009). The term $1/(1+\beta)$ in Eq. (4) represents a traditional branching ratio (i.e., $k_{R6b}/(k_{R6a}+ k_{R6b}) = [HNO_3]/([NO_2]+[HNO_3])$). For the experiments conducted, the value of $\beta$ is at most 0.04, leading to a 2 % correction to equation 1. See Sect. S2 of the supplement for information regarding the formulation of Eq. (4) and the calculation of $\beta$.

The CIMS response to HONO is determined by acquiring a background by briefly toggling off the mercury lamp. This background CIMS signal is humidity dependent, so a background is taken at each humidity setting. Background CIMS I(HONO)$^-$ signals are elevated during calibrations due to impurities in the NO flow.

**2.2.2 Proxy Calibration**

For the $NO_2$ proxy calibration method, we determine [HONO] from the measured value of [$NO_2$] formed from R6a during HONO production. For each $H_2O$ molecule photolyzed (R4), nearly one $NO_2$ and two HONO molecules are produced. Therefore, [HONO] is simply given by the measured [NO2] (Eq. (5)):

$$[HONO] = (2 + \beta) \cdot [NO_2] \tag{5}$$

where $\beta$ is added to account for the minor $HNO_3$ product of the $HO_2 + NO$ reaction (R6b).

For the proxy calibration we use humidified $N_2$ rather than air and include a small addition of 10 sccm of zero air prior to lamp exposure (not shown in the Fig. 1 schematic). The resulting low $O_2$ concentration ([$O_2$] = 0.040 ± 0.002 %) is sufficient for the full conversion of H to $HO_2$ (R5) but results in a negligible amount of $O_3$ formed by $O_2$ photolysis (R7-R8), confirmed by toggling the UV source on and off with dry carrier gas flowing. The pseudo-first order rate constant for the H to $HO_2$ conversion (R5) is $1.1 \times 10^4$ $s^{-1}$ for this [$O_2$] value. HONO concentrations are quantified using background subtracted [$NO_2$] values in Eq. (5), which are typically acquired by toggling the mercury lamp off and on. The CIMS signal response is determined simultaneously. A "direct" $NO_2$ detection method is highly recommended over indirect methods that rely on NO detection as the high NO mixing ratios would result in degraded precision. For this study we used a CAPS instrument but other methods like cavity ring down spectroscopy (CRDS), laser-induced fluorescence (LIF), or oxygen anion CIMS (e.g., Novak et al. (2020)) would be acceptable.

**3 Results and Discussion**

Time series data for the proxy calibration method are shown in Fig. 2. The CAPS $NO_2$ measurement and the CIMS HONO signal normalized to one million counts per second of reagent ion ('ncps') are shown at a constant humidity (RH = 29 %, $\chi_{H2O}$ = 0.0065, T = 19.3 °C, P = 760 Torr within the photolysis cell and RH = 19 %, $\chi_{H2O}$ = 0.0042, T = 19.2 °C, P = 760 Torr within the CIMS IMR exhaust). During the first 120 s shown in Fig. 2, HONO is formed by $H_2O$ photolysis via the mercury lamp 184.9 nm emission. This leads to the stable I(HONO)$^-$ signal in the CIMS along with enhanced [$NO_2$] produced by R6a and measured by the CAPS monitor. Background I(HONO)$^-$ and $NO_2$ signals are determined by toggling off the mercury lamp (shown at 121 s). The CIMS sensitivity (ncps $ppt^{-1}$) is equal to the quotient of the normalized background subtracted CIMS signal and the quantified [HONO] which is calculated by Eq. (5).

A multipoint $NO_2$ proxy calibration curve (Fig. 3) shows the linear CIMS signal response to [HONO]. This calibration was conducted at a constant relative humidity (RH = 28 %, $\chi_{H2O}$ = $6.22 \times 10^{-3}$, T = 19.3 °C, P = 760 Torr within the photolysis cell and RH = 18 %, $\chi_{H2O}$ = $3.88 \times 10^{-3}$, T = 19.2 °C, P = 760 Torr within the CIMS IMR exhaust), and [HONO] was adjusted by altering the mercury lamp flux with a Variac variable transformer. The slope of this curve, 2.89 ± 0.34 ncps $ppt^{-1}$ (2σ), is the CIMS sensitivity to HONO for this particular water mole fraction within the CIMS IMR ($\chi_{H2O}$ = $3.88 \times 10^{-3}$). HONO mixing ratios ranged from approximately 400 pptv to 3500 pptv, thus demonstrating that a wide range in HONO concentrations can easily be prepared. The uncertainties for the quantified [HONO] values from Eq. (5) (i.e., the x-error bars) are obtained by adding in quadrature three terms: 1.) the relative uncertainty of the $NO_2$ background subtraction (based on the 5 s average precision of 27 pptv), 2.) the $NO_2$ calibration uncertainty (3%, 2σ), and finally 3.) the relative uncertainty associated with the (2 + $\beta$) expression (typically 0.14%, 2σ). The uncertainty in the normalized CIMS signal is obtained by adding in quadrature the 15 s precision of the I(HONO)$^-$ signal with that of the reagent ion. The 2σ error bars range from 6.1 to 34.8 % for quantified [HONO] and 1.7 to 2.9 % in the CIMS HONO signal. The Fig. 3 slope (i.e., the CIMS sensitivity to HONO) and its uncertainty were determined using the York bivariate

linear regression method (York et al., 2004). The uncertainty calculations are discussed in greater detail in Sect. S4.1 of the supplement.

A comparison between the more standard $O_3$ actinometry based calibration and the new proxy calibration method is shown in Fig. 4. CIMS sensitivities as determined by single point calibrations are shown for a variety of $\chi_{H2O}$ values. The two calibration methods were conducted consecutively and agree within their provided $2\sigma$ errors. Sensitivities ranged from 1.5 to 5.3 ncps ppt$^{-1}$ with the greatest values observed at low $\chi_{H2O}$ settings. The CIMS IMR $\chi_{H2O}$ values ranged from $1.77 \times 10^{-3}$ to $8.25 \times 10^{-3}$ and corresponded to a photolysis cell RH range of 4.1 to 71 % and photolysis cell $\chi_{H2O}$ values of $0.93 \times 10^{-3}$ to $16 \times 10^{-3}$ (average T = 19.6 $^o$C; P = 760 Torr). The sensitivity determined by the Fig. 3 multipoint calibration (2.89 $\pm$0.34 ncps ppt$^{-1}$ at CIMS IMR $\chi_{H2O}$= $3.88 \times 10^{-3}$) is consistent with those shown in Fig. 4 at similar $\chi_{H2O}$ values. These CIMS sensitivities are also in line with literature values (Peng et al., 2020; Bourgeois et al., in review, 2022). Unique to this figure is the use of the CIMS IMR $\chi_{H2O}$ to track humidity dependence rather than the partial pressure of $H_2O$ (Lee et al., 2014), the specific humidity (Novak et al., 2020), or the CIMS signal ratio of the iodide water adduct $I(H_2O)^-$ (m/z 145) to reagent ion $I^-$ (m/z 127) (Lee et al., 2014; Peng et al., 2020; Veres et al., 2015; Veres et al., 2020). The use of $\chi_{H2O}$ allows for a more direct comparison to other CIMS instruments that may use different IMR pressures and quadrupole voltage settings that govern the $I(H_2O)^-$ to $I^-$ ratio. See the supplement for more information regarding humidity effects on ionization chemistry (Sect. S1.2).

The uncertainty in the CIMS sensitivity for the single-point proxy calibration points in Fig. 4 is determined by combining in quadrature the relative uncertainty of the background subtracted CIMS signal with that of the quantified [HONO] value. The calculation for quantified [HONO] uncertainty and CIMS uncertainty is previously mentioned in the text regarding Fig. 3, though the CIMS uncertainty here slightly differs due to the background subtraction (see Sect. S4.2 the supplement). As mentioned previously, the total uncertainty depends on the quantified [HONO] value, which ranged from 630 (at the lowest $\chi_{H2O}$ value) to 7,800 pptv (at the greatest $\chi_{H2O}$ value) for the proxy calibration results shown in Fig. 4. The uncertainty for the greatest [HONO] values (corresponding to the highest $\chi_{H2O}$ settings) are dominated by the 4 % $2\sigma$ uncertainty of the $NO_2$ measurement. At the lowest [HONO] values (shown at lowest $\chi_{H2O}$ settings), larger uncertainties occur and are dominated by the precision in $NO_2$ measurements due to the background subtraction. Therefore, the uncertainty can be minimized by using higher HONO concentrations. The presented $2\sigma$ uncertainty in sensitivity using the proxy method ranged from 5.1 to 25.5 % with the [HONO] quantification alone accounting (from E.q. (4)) for 4.5 to 24.4 %. These proxy calibration uncertainties for [HONO] fall well below the 27 % $2\sigma$ uncertainty associated with the standard $O_3$ actinometry calibration. The $2\sigma$ uncertainty in sensitivities determined by our actinometry calibrations (i.e., the 27 % method uncertainty combined with precision of the CIMS measurement) varies from 27.1 to 28.8 %. The proxy calibration therefore allows for lower uncertainties compared to the standard actinometry calibration, especially at high [HONO] values.

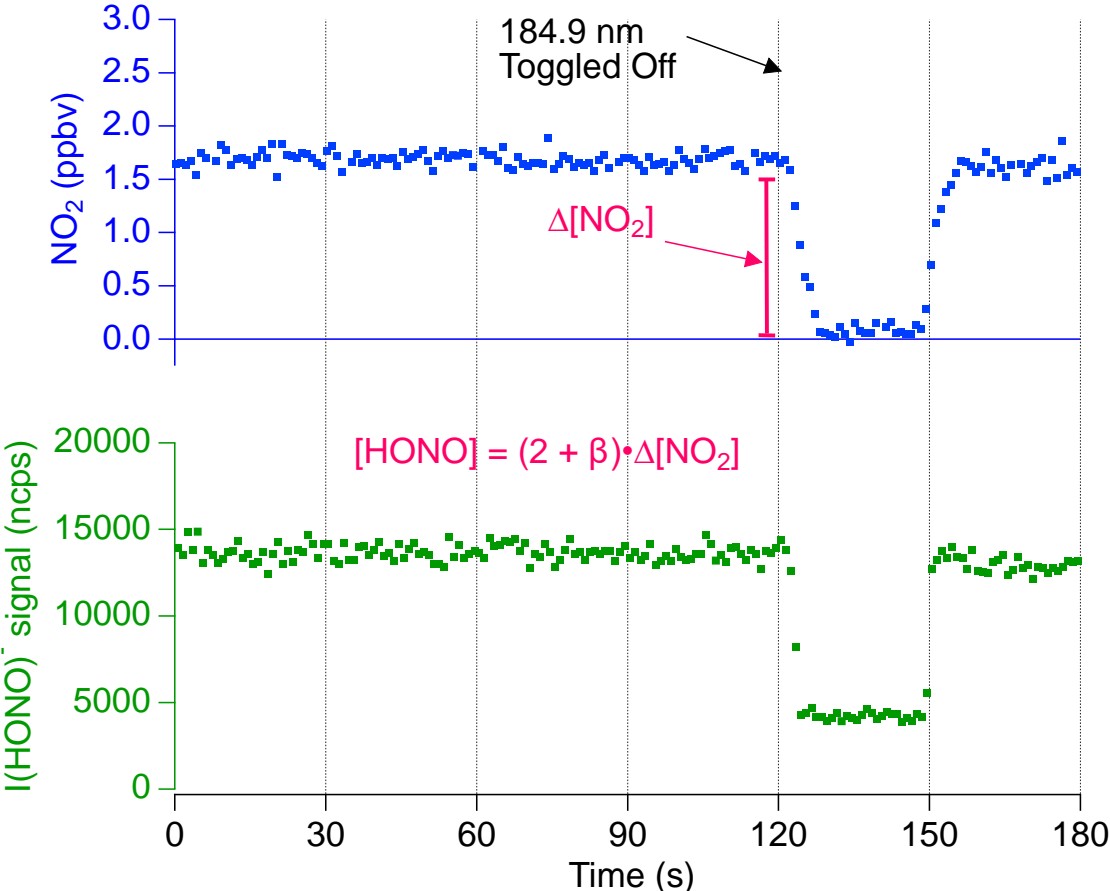

**Figure 2: One-second averaged time series data for a proxy calibration at a constant relative humidity. The iodide HONO adduct signal (cps) is shown normalized per one million reagent ions (ncps). The shaded section represents the period in which the 184.9 nm mercury lamp is toggled off to obtain background [NO₂] and HONO signal. The NO₂ concentration is shown with an offset so that background values are near 0 ppbv. The resulting difference in NO₂ indicates that approximately 3,000 pptv [HONO] is sampled by the CIMS.**

275

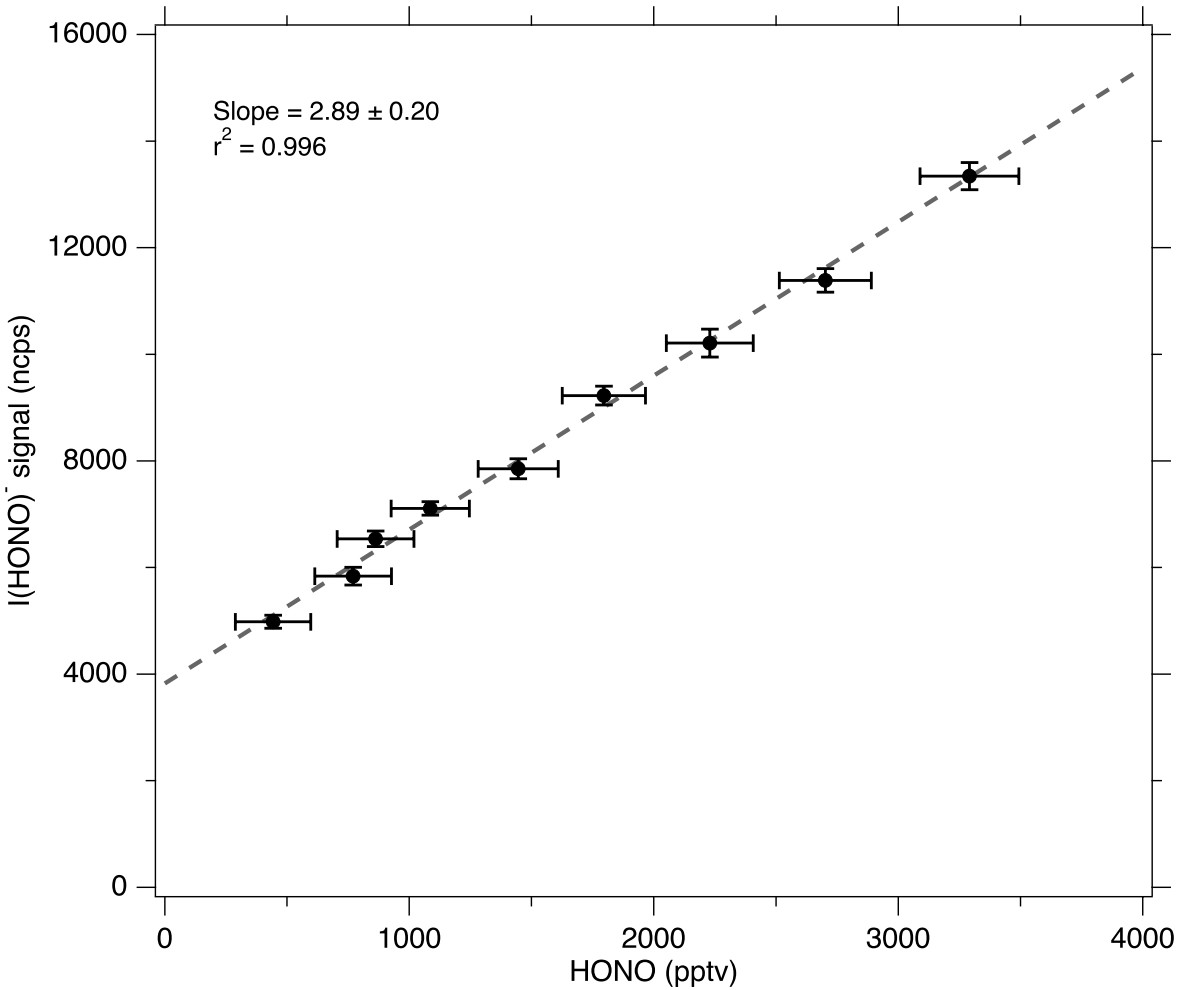

Slope = 2.89 ± 0.20
$r^2 = 0.996$

280

**Figure 3: Calibration curve obtained using the NO₂ proxy calibration method at a constant humidity (RH = 17.7%, [H₂O] = 0.388% as measured in the CIMS scroll pump exhaust) by varying photon flux *F* with a Variac variable transformer. Error bars represent ± 2σ uncertainty. The data was fitted using the York bivariate regression method (York et al., 2004). The y-intercept (3,820 ncps) represents the background CIMS I(HONO)⁻ signal during this calibration and is mostly from impurities in the constant NO addition. Without the**

285 **NO addition (for ambient sampling) the typical background signal is 75 ncps.**

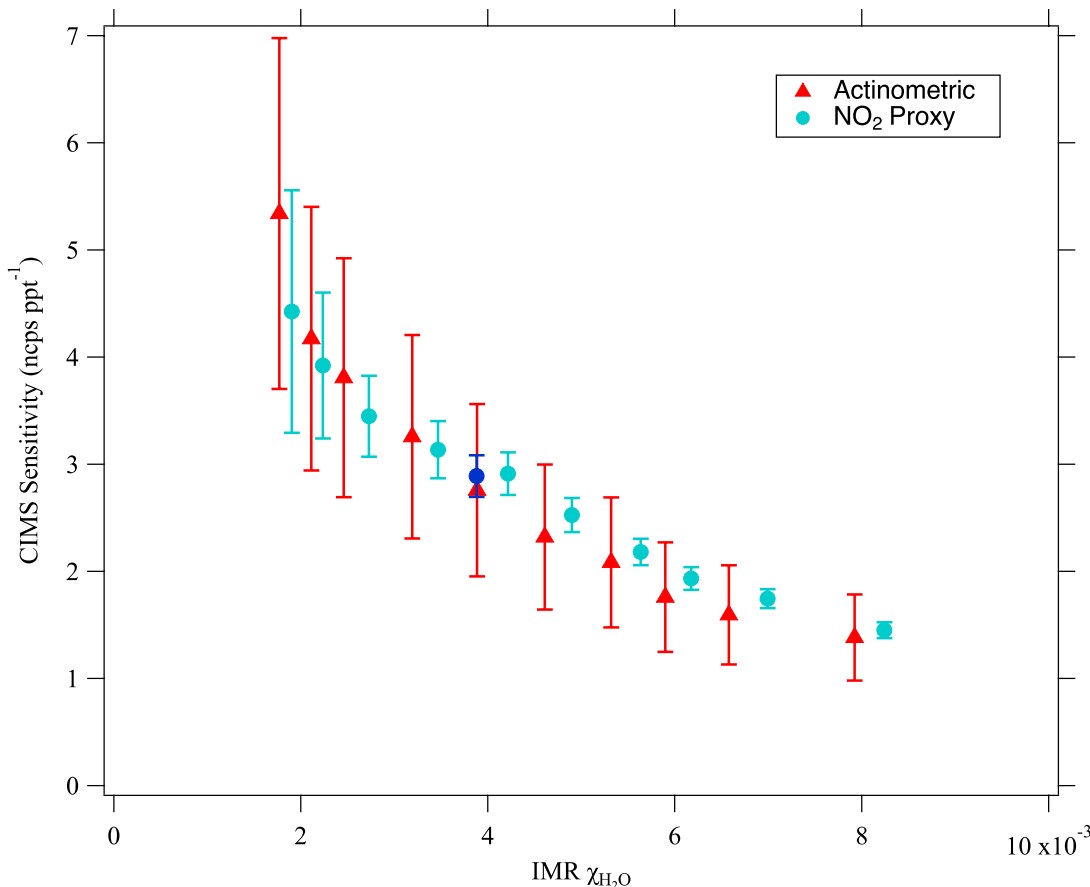

**Figure 4: Comparison of the two HONO calibration methods for a range of $\chi_{H2O}$ within the CIMS IMR. The sensitivity determined by the multipoint calibration (i.e., the Fig. 3 slope) is plotted as the dark blue circle. Error bars represent ± 2 $\sigma$ uncertainties. [HONO] ranged from 470 pptv (at the lowest $\chi_{H2O}$ value) to 7,710 pptv (at the greatest $\chi_{H2O}$ value) for the actinometric calibration. [HONO] similarly ranged 630 pptv to 7,820 pptv for the NO₂ proxy calibration.**

## 4 Application to HOx Calibrations

In addition to its use described herein for HONO calibrations, the NO$_2$ proxy method can also be used to determine the prepared HOx concentration (i.e., prior to its reaction with NO), which differs from [HONO] by only a few percent depending on the value of $\beta$.

$$[HO_x] = (2 + 2\beta) \cdot [NO_2] \tag{6}$$

Directly using this method to calibrate a HOx instrument (e.g., LIF, HOx-CIMS, or perCIMS) is likely not feasible as the described calibration is performed in N$_2$ rather than air (i.e., the composition of the calibration carrier gas should be identical to that for ambient air) and requires high NO mixing ratios that could complicate the operation of a HOx instrument more than it does for our iodide CIMS.

The application of an NO$_2$ proxy calibration for a HOx instrument would likely require two consecutive steps. First an NO$_2$ proxy calibration would be performed as described in this manuscript (with high [N$_2$], high [NO], and only the NO$_2$ instrument sampling). Second, the N$_2$ and NO flows would be replaced with zero air, and the resulting calibration mixture, which would have the same HOx concentration as determined by the NO$_2$ proxy calibration, would then be sampled by a HOx instrument. Additional concentrations of HOx could be prepared by altering and tracking [H$_2$O]:

$$[\text{HOx}] = \frac{[\text{HOx}]_{proxy}}{[\text{H}_2\text{O}]_{proxy}}[\text{H}_2\text{O}] \tag{7}$$

where the 'proxy' designation refers to the corresponding values of [HOx] (quantified via Eq. 6) and [$\text{H}_2\text{O}$] during the proxy calibration step (i.e., high [NO] and $\text{N}_2$ as the carrier gas). The quotient of [HOx] and [$\text{H}_2\text{O}$] is equal to the product of the constants $F \cdot t \cdot \sigma_{H2O} \cdot \Phi_{HOx}$ that appear in Eq. 1.

The uncertainty in [HOx] will likely range 5 to 15 % ($2\sigma$), similar to that for [HONO] quantified via proxy calibration, mainly dependent on the [$\text{NO}_2$] measurement uncertainty. While the accuracy in $\text{H}_2\text{O}$ measurements should not contribute to the Eq. 7 [HOx] uncertainty as it effectively cancels out, the precision in the [$\text{H}_2\text{O}$] measurements would have to be accounted for. The uncertainty for this proposed method is exceptional compared to typical HOx calibrations in which total uncertainties are often above 25 % ($2\sigma$) based on the combination of Eq. (1) parameters of $F \cdot t$, $\sigma_{H2O}$, and [$\text{H}_2\text{O}$].

## 5 Conclusions

Two photolytic HONO calibration methods based on reacting NO with the HOx generated by $\text{H}_2\text{O}$ photolysis at 184.9 nm were presented. This includes a novel approach in which HONO is quantified using the $\text{NO}_2$ formed by the $\text{HO}_2$ + NO reaction as a proxy. The proxy method compares well with the $\text{O}_3$ actinometry based calibration while also having the benefit of a simpler calculation that avoids the need to characterize the emission spectrum of the mercury lamp used. In addition, this proxy method has improved uncertainties, typically between 4.5 and 10 % ($2\sigma$) – lower than the 27 % $2\sigma$ uncertainty associated with the actinometric calibration method. We also detail the potential application of a $\text{NO}_2$ proxy calibration for HOx calibrations, in which we anticipate exceptional $2\sigma$ uncertainties of below 15 %. These photolytic calibrations require a direct $\text{NO}_2$ measurement, a 184.9 nm light source, and a simple quartz tube photolysis chamber. While the proxy calibration method was conducted in $\text{N}_2$ for this manuscript, it is possible to instead perform this method using air in the case that $\text{N}_2$ is incompatible with an instrument (unlike our CIMS). In an air-based proxy calibration, the [$\text{O}_3$] produced by $\text{O}_2$ photolysis would need to be quantified (during dry conditions) and then subtracted from subsequent background subtracted $\text{NO}_2$ signals, but at the expense of greater calibration uncertainty. In conclusion, these photolytic calibration techniques offer a valuable alternative to the more conventional HONO calibration that is based on reacting hydrogen chloride vapor with sodium nitrite.

*Data availability.* The data used in this manuscript is available upon request.

*Author contributions.* AJL and ECW designed the experiments. AJL carried them out and performed the data analysis. AJL prepared the manuscript with contributions from ECW.

*Competing Interests.* The authors declare that they have no conflict of interest.

*Acknowledgements.* We acknowledge financial support from Directorate for Geosciences of the National Science Foundation. This work was supported by NSF grant AGS-2002928. We are grateful to Phil Stevens (Indiana University) for characterizing the mercury lamp used for these calibrations.

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
