# Peer review of "Comparison of Two Photolytic Calibration Methods for Nitrous Acid"

_Atmospheric Measurement Techniques, 2022_

## Author Comment (AC1)

**Response to public comments made by Jörg Kleffmann.**

**Comparison of Two Photolytic Calibration Methods for Nitrous Acid**

**Prepared by Andrew J. Lindsay and Ezra C. Wood**

**We are thankful for the public comments made by Dr. Kleffmann during the interactive discussion phase. Black text denotes comments made by Dr. Kleffmann, and blue text denotes our responses.**

In the manuscript of Lindsay and Wood a new quantification method used for a former photolytic HONO source is described. In the source, HONO is formed by photolysis of water at 184 nm forming OH and by the consecutive reaction of NO+OH. HONO is quantified by measuring the additional reaction product $NO_2$ ("$NO_2$ proxy method"). In addition, to several comments by the three reviewers, I have also a few other comments to the manuscript.

In the introduction, I missed a short summary on other HONO sources used in former studies besides the Febo et al. source and the photolytic sources. First, there are recent modifications of the Febo source and second, also other types of HONO sources are completely missing (e.g. the one by Taira and Kanda, 1990 or the very recent one from our group, Villena and Kleffmann, 2022). In addition, in contrast to the statement by the authors in lines 54-56, the original Febo source can be operated down to a few ppbs (see the original publication) and in recent modifications of this source, HONO levels even in the sub-ppb range can be produced.

The Villena and Kleffmann, 2022 paper was cited within the fourth paragraph of the preprint's introduction focused on recent modifications to the Febo technique. We will add additional text regarding the ability of Febo-based sources to operate at lower concentrations. Specifically, we will reference a recent modification by Lao et al., 2020 that can operate down to tens of pptv.

In addition, the authors should highlight that their HONO source represents a complex $NO_y$ mixture including NO (in excess), $NO_2$ (50% of HONO), HONO and $HNO_3$ and is not a more or less pure HONO source like in most former approaches (e.g. the purity of HONO from the original Febo source was >99%). This makes the use and quantification of this source more complicated.

Although the HONO calibration mixture does include $NO_2$, excess NO, and trace $HNO_3$, we respectfully disagree that this makes the calibration gas "complex". The NO, $NO_2$, and $HNO_3$ do not add any complexity to our CIMS measurements besides the higher background signal. We will clarify in the revision that the high NO could potentially preclude its use for other HONO measurement methods (while noting that it worked very well for iodide-adduct CIMS). We disagree that the use and quantification of

our source is complicated – in fact, as presented in the manuscript, the quantification of HONO using the proxy method is actually very simple!

For example, the absolute interferent-free quantification of $NO_2$ is absolutely necessary for the present approach, which is not trivial here. E.g. the typical chemiluminescence instruments with molybdenum converters ("NO-what-boxes") commonly used for the simply quantification of pure HONO sources cannot be used here. And even if a more selective photolytic converter is available, the quantification of $NO_2$ is highly uncertain, since a) there is the additional uncertainty in the $NO_2$-converter efficiency and b) $NO_2$ is quantified from the difference of two large signals (NO is in excess...). Thus, groups who want to use this source need to have a CAPS or any similar selective and direct $NO_2$ instrument. In addition, in this humid $NO_y$ mixture, there may be significant secondary heterogeneous HONO formation ($NO+NO_2+H2O$, $2NO_2+H_2O$, heterogeneous photolytic $NO_2$ conversion…), which is dependent on the surfaces available (photoreactor, transfer lines, analyzer,…), the gas/surface reaction time and S/V ratio and which will affect both, the concentrations of HONO and of $NO_2$ used to quantify HONO.

We completely agree with these points regarding the need to use a direct $NO_2$ measurement (e.g., CAPS or CRDS), and this was already noted within the conclusion of the preprint. In the revision we will include additional mentions of this requirement in the introduction and abstract.

Non-photolytic heterogeneous HONO formation mechanisms are accounted for during the calibration procedure by obtaining the background signal by turning off and on the mercury lamp (i.e., any heterogeneous formed HONO will be present during both background and operation). In the revision we will discuss the possibility that additional HONO could be formed by exposure of the quartz photolysis tube to the UV radiation or heterogenous formation of HONO through photolytic $NO_2$ conversion. The results of several tests, some already conducted, will be included (e.g., turning the UV on/off without the excess NO flow under both dry and humid conditions, and turning a 254 nm-only source on/off with humidified $N_2$/air and excess NO).

Furthermore, we note that the alternative to using a humidified calibration gas is to only calibrate an instrument under dry conditions. The response of any HONO instrument to ambient humidity must be thoroughly characterized, so these issues regarding potential heterogeneous HONO formation may also apply to other HONO measurement methods.

Besides, the authors should specify the range of HONO levels, which can be obtained by the independent variation of the three variables (light intensity, humidity, reaction time). This is important, since for example the variation of the humidity may not be recommended when calibrating a CIMS instrument, caused by the strong, non-linear humidity dependence of these instruments (see Figure 4).

Different values of HONO can be prepared by varying light exposure, light intensity, and humidity. The multipoint calibration plot (Fig. 3) was obtained at a constant RH with HONO varied by altering photon flux (i.e., light intensity). There is a factor of seven difference between the greatest and smallest [HONO] values obtained. The caption of this figure will be updated to state experimental conditions (i.e., constant humidity with photon flux altered using a Variac Variable transformer) so that readers can get an idea of a potential concentration range by adjusting photon flux only.

The [HONO] values of Fig. 4 (i.e., the sensitivity-humidity plot) were adjusted by altering humidity only. We had a factor of 15 difference between the highest and lowest [HONO] values that were obtained by varying humidity between 4.1 % and 67 % (RH values within the photolysis tube). There is a proportional

relationship between [HOx] and [H₂O] and a near-proportional relationship between [HONO] and [H₂O] at constant lamp flux and exposure (i.e., both relative humidity and [HONO] were increased nearly 15 fold). In the revision of the text, the range in [HONO] and the photolysis tube humidity (both RH and mixing ratios) will be stated in the text to make the potential range in [HONO] by varying only humidity more apparent. The experimental conditions for Fig. 3 (the multipoint calibration) and Fig. 4 are both stated in the text.

**Specific comments:**

Line 31: Should be Jiang et al., 2020 (no 2022 paper in the reference list?)

Lines 88-90: Can you explain how the humidity dependence is accounted for the CIMS? This should be a non-linear correction, see Figure 4, the shape of which may be in addition HONO dependent (with decreasing sensitivity at high HONO levels (?) as this was observed for the CIMS used in the study of Jurkat et al., 2011, doi:10.1029/2011GL046884).

In addition, can the instrument's analytical parameters by specified (DL, precision, accuracy, linear range), see the variable signal background in Figures 2 and 3 and the significant noise at the 5 ppb HONO level in Figure 2.

**Response to Specific Comments:**

Reference issue: Thank you for noticing this issue. The Jiang et al., 2022 citation is correct but was missing from the references section.

Accounting for humidity: For ambient sampling, we constantly measure humidity and apply a function to determine the real-time CIMS sensitivity (similar to how almost all iodide-CIMS users operate). We use an exponential fit to describe the relationship between I-HONO⁻ signal and CIMS IMR $\chi_{H2O}$. We will add a sentence briefly outlining this procedure, but full details on our ambient measurements will be discussed an upcoming manuscript. We have not observed any decreases in sensitivity with [HONO].

Analytical parameters: We will include the below information regarding detection limit, precision, and linear response in the SI as this manuscript is focused on calibration techniques and not instrumental parameters.

- **DL:** We do not state or show a true HONO background signal within this paper (i.e., HONO signals were elevated from impurities in our NO cylinder). During sampling near Boise, Idaho, background I-HONO⁻ counts were ~190 normalized counts per second (ncps). This led to a detection limit of 12.3 pptv for HONO at SNR = 2 (time averaging of 1 s).

- **Precision:** shot noise dominated (i.e., the square root of the signal in counts/s)

- **Accuracy:** The accuracy depends on the calibration method, which is one of the main points of this paper.

- **Linear range:** We show linear response using the multipoint calibration for a concentration range of 450 to 3,400 pptv. We have yet to discover a limit to the linear range.

- **Inconsistent background signal in Figures 2 and 3:** The time series figure (Fig. 2) was of an early experiment and used insufficient [NO] (low by a factor ~ 4). The time series figure has been updated for the revised version of the manuscript, and the background signal is now consistent with the Fig. 3 multipoint calibration intercept.

- **Figure 2 Noise:** The Fig. 2 time series plot shows 1 Hz data and we disagree that it shows "significant" noise for this concentration ~ 5 ppbv HONO. 60-sec averages would show markedly reduced noise.

---

## Author Response (AR2)

**Reviewer Response Document.**

**Comparison of Two Photolytic Calibration Methods for Nitrous Acid**

**Prepared by Andrew J. Lindsay and Ezra C. Wood**

This initial page (page 0) was prepared in response to technical corrections suggested by the associate editor. The responses to comments made by anonymous reviewers begins on page 1.

We thank our editor for making the following suggestions. Responses are indicated with blue text, and green text represents the text within the revised document.

**Technical corrections:**
On pg 6, line 180: '14.9 nm' should be changed to '184.9 nm'.

Thank you for noticing this typo. It has been fixed accordingly.

In the updated fig 2 caption you state that 'the $NO_2$ concentration is shown with an offset so that background values are near 0 ppb'. I think it would be useful if you could add a comment somewhere in the manuscript on what causes this background signal in the $NO_2$ CAPS?

A comment in reference to the offset background has been added to the results and discussion section focused on the Fig. 2 time series. The updated text is included below. New text is shown highlighted.

**3 Results and Discussion**

Time series data for the proxy calibration method are shown in Fig. 2. The CAPS $NO_2$ measurement and the CIMS HONO signal normalized to one million counts per second of reagent ion ('ncps') are shown at a constant humidity (RH = 29 %, $\chi_{H2O}$ = 0.0065, T = 19.3 °C, P = 760 Torr within the photolysis cell and RH = 19 %, $\chi_{H2O}$ = 0.0042, T = 19.2 °C, P = 760 Torr within the CIMS IMR exhaust). The $NO_2$ data is shown with an offset so that concentrations are near 0 ppbv while the UV source is off. Background [$NO_2$] values were typically near 15 ppbv due to impurities within the NO cylinder and $NO_2$ production from the termolecular reaction $NO + NO + O_2 \rightarrow 2\ NO_2$. During the first 120 s shown in Fig. 2, HONO is formed by $H_2O$ photolysis via the mercury lamp 184.9 nm emission. This leads to the stable $I(HONO)^-$ signal in the CIMS along with enhanced [$NO_2$] produced by R6a and measured by the CAPS monitor. Background $I(HONO)^-$ and $NO_2$ signals are determined by toggling off the mercury lamp (shown at 121 s). The CIMS sensitivity (ncps ppt$^{-1}$) is equal to the quotient of the normalized background subtracted CIMS signal and the quantified [HONO] which is calculated by Eq. (5).

**Reviewer Response Document.**

**Comparison of Two Photolytic Calibration Methods for Nitrous Acid**

**Prepared by Andrew J. Lindsay and Ezra C. Wood**

We thank our 3 anonymous reviewers for spending the time to look over our manuscript and provide meaningful feedback. We are also thankful for the public comments made during the interactive discussion phase. We believe the resulting revisions have improved this manuscript.

In this document, we respond point by point to each reviewer and commenter.

**The responses to reviewers #1, #2, and #3 are provided on pages 2, 12, and 15, respectively.**

Besides the edits made in response to reviewer comments, we have made additional changes that we detail here:

- **We have included a new section discussing possible application of the proxy calibration method to HOx calibrations.** Briefly, the [NO$_2$] produced via HO$_2$ reacting with NO can be related back to the [HOx], and thus to $F \cdot t \cdot \sigma_{H2O}$ (the product of the UV photon flux, the irradiance time, and the water vapor absorption cross section). Once $F \cdot t \cdot \sigma_{H2O}$ is determined in N$_2$ with high [NO], the flow within the photolysis tube can be replaced with humidified zero air for calibrating a HOx instrument (e.g., LIF). Values of [HOx] can be tuned by adjusting the humidity.

- **We now include a supplement section that tabulates the contributions to the 27 % combined uncertainty that we use for the actinometric calibration.**

We would also like to note changes to figures presented within the manuscript.
- **The time series figure (Fig. 2) showed data from an outdated experiment that used a lower value of [NO].** We have updated the figure in response to anonymous reviewer # 3 who noticed a significant difference in the presented CIMS background signal compared to the other data shared. This reviewer also attempted to calculate CIMS sensitivity from this figure and found deviations to the results shown in Fig. 4. Briefly, the low [NO] value of the original time series was insufficient for the conversion of OH to HONO by reaction with NO (e.g., due to the competitive reaction of OH with sample tubing surfaces) but sufficient for the reaction of HO$_2$ with NO, forming NO$_2$. See the relevant responses to anonymous reviewer # 3 on pages 17 and 18

- **The error bars in the sensitivity-humidity figure (Fig. 4) were incorrect and have been updated.** The "y-error" bars for the NO$_2$ proxy calibration were erroneously based on the precision of the CIMS 1-sec data while the standard actinometric method did not include CIMS noise. The y-error bars now include a contribution from the precision of 15-sec average CIMS data for both calibration types. The result is improved (lower) sensitivity uncertainty for the NO$_2$ proxy calibration. The relative 15 sec CIMS precision is at most 3 % (2σ).

**Point-by-Point Response to Anonymous Reviewer 1:**

**Abstract.**

- State the range of HONO concentrations over which the calibration methods operate
  - We have updated the abstract accordingly. We now state, "both calibration methods were used to prepare a wide range in [HONO] of ~400 to 8,000 pptv for both calibration methods."

  - We also have included within the abstract a greater discussion of the calibration uncertainties and how uncertainty varies with HONO concentration. The new abstract text is provided below:
    The uncertainty of the chemical actinometric calibration is 27 % (2σ) and independent of HONO concentration. The uncertainty of the $NO_2$ proxy calibration is concentration-dependent, limited by the uncertainty of the $NO_2$ measurements. The $NO_2$ proxy calibration uncertainties (2σ) presented here range from 4.5 to 24.4 % (at [HONO] = 8,000 pptv and [HONO] = 630 pptv, respectively) with a 10 % uncertainty associated with a mixing ratio of ~1,600 pptv, typical of values observed in urban areas at night. We also describe the potential application of the $NO_2$ proxy method to calibrating HOx instruments (e.g., LIF, CIMS) at uncertainties below 15 % (2σ).

**Introduction:**
- HONO is very important indoors, given the wavelength cut off for window glass, and mention of HONO measurements indoors should be made
  - The highlighted sentence in the paragraph below has been added to reflect the importance of HONO indoors.
    "HONO photolysis has been reported as a major source of HOx (HOx = OH + $HO_2$) throughout the day in a variety of environments, including urban and highly polluted areas (Whalley et al., 2018;Slater et al., 2020;Ren et al., 2013;Lu et al., 2019) as well as more pristine environments (Villena et al., 2011;Jiang et al., 2020;Bloss et al., 2007). Vertical distributions of HONO, however, indicate that its significance as a HOx precursor may be limited to near ground level (Li et al., 2014;Young et al., 2012;Villena et al., 2011;Wong et al., 2012;Tuite et al., 2021;Jaeglé et al., 2018). HONO can also serve as an important source of HOx in indoor environments since sufficient UV light can penetrate windows and substantial HONO concentrations can result from various activities (e.g., cooking) (Gomez Alvarez et al., 2013; Wang et al., 2020)."

- Page 2, line 40. Some methods used to detect HONO are listed in this paragraph, with a focus on intercomparisons. One method not listed is that of laser photofragmentation followed by detection of OH using laser-induced fluorescence spectroscopy, this ought to be listed having been used for both indoor and outdoor HONO measurements.
  - This particular section of the introduction is focused on intercomparison studies and is not meant to present an overview of HONO measurement techniques. We have, however, mentioned photofragmentation at a later portion of the introduction. This technique was used for two of the three previously conducted literature photolytic-based calibrations. The updated text is shown highlighted.

    The HONO formed has been quantified based on the water vapor mixing ratio, water vapor absorption cross section, the UV flux, and the UV exposure time. This quantification approach thus far has been used to calibrate HONO photo-fragmentation instruments that detect OH using laser-induced fluorescence spectroscopy (Dyson et al., 2021; Bottorff et al., 2021).

**Instrumentation section:**
- Page 3, line 75, state the range of O3 over which the CAPS instrument was calibrated

- The range in [$O_3$] was 10 to 300 ppbv. This has been included in the text along with additional information regarding the calibration technique of titrating $O_3$ with excess NO requested by reviewer #3. Relevant text from the revised manuscript is included below, and the new text is shown highlighted.

  A Cavity Attenuated Phase Shift (CAPS) spectrometer (Aerodyne Research, Inc.) was used to detect $NO_2$ (Kebabian et al., 2008). The CAPS also indirectly measured O3 as it was converted to $NO_2$ by reaction with excess NO. The CAPS instrument was calibrated using a 2B Technologies Model 306 $O_3$ Calibration Source. Ozone outputs were varied between 10 and 300 ppbv with greater than 99.99 % conversion efficiency to $NO_2$ by reaction with excess NO ([NO] = 1.82 ppmv) within approximately 15 m of FEP tubing (i.d. = 0.476 cm; residence time = 17.1 s, pseudo-first order rate constant of 0.8 $s^{-1}$). The manufacturer stated accuracy of this $O_3$ calibrator is 2 % ($2\sigma$), though no recent factory calibrations have been conducted. Therefore, a second calibration was conducted with a Thermo Environmental Instruments 49C $O_3$ Calibrator, which agreed to within 2.5 %. We assign an uncertainty of 4 % ($2\sigma$) to the $NO_2$ measurements to account for possible drift in accuracy.

- Although the manufacturer for the NO gas used is stated on page 4 (Airgas), manufacturers for other gases are not given, and none of the purities of the gases are stated, nor of the purity of the water used in the bubbler (electrical resistance) or other reagents (e.g. CH3I). Given HONO is such a difficult molecule to measure, with impurity and interference problems, it is important to state the purity of the gases/reagents used in the calibration.
  - The manufacturer and purity grade for all gases/gas mixtures used in this manuscript are now stated. We find that including gas specifications is cumbersome within the text. For example, the "ultra zero" grade air includes possible impurities of total hydrocarbons and CO of less than 0.1 and 0.5 ppmv, respectively. We use a lower grade purity of $N_2$ (Airgas, "industrial" grade), though we have found no differences when comparing to results acquired using ultra high purity $N_2$. For reference, $N_2$ is used for the proxy calibration carrier gas and is also constantly used for CIMS sampling. We have included a statement in the text to justify the use of low purity $N_2$ here.

    Updated text regarding gases (and water) is shown highlighted here:
    We calibrate HONO using two variations of the water vapor HOx calibration method: one is a modification of the standard actinometric HOx photolytic calibration and the other we refer to as the "$NO_2$ proxy" calibration. These calibration methods mainly differ in how HONO is quantified. In both methods, HONO was produced nearly identically. Air (Airgas, Ultra Zero grade) for the actinometric method or $N_2$ (Airgas, industrial grade) for the $NO_2$ proxy calibration is humidified with HPLC (high-performance liquid chromatography) grade water (Fisher Chemical), mixed with NO (Airgas, 41.02 ± 2.05 ppmv in $N_2$), and then exposed to 184.9 nm ultraviolet radiation from a low-pressure mercury lamp (Jelight 78-2046-1). While in this manuscript we use industrial grade $N_2$ for both the proxy calibration carrier gas and the humidified $N_2$ CIMS IMR input, we have used ultra high purity $N_2$ (Airgas) in previous experiments. We find no differences between the calibration results acquired using different grades of $N_2$. The resulting OH and $HO_2$ from water photolysis form HONO by reaction with excess NO (R4-6a and R2).

    The $CH_3I$ permeation tube manufacturer and relevant information is now included (updated text shown here: The $I^-$ was prepared by exposing dilute methyl iodide ($CH_3I$) from a permeation tube (VICI Metronics) to a $^{210}$Po radioactive source).

  - The purity of water was not assessed by conductivity measurements. We used fresh HPLC grade water.

- State the manufacturer of the RH/ T probe. Quantification of water vapour is a central part of the calibration method, and water vapour is difficult to measure accurately. Was a dew-point hygrometer or other instrument for water vapour used, even if only to check the calibration of the RH probe (as these can drift).
  - We used two Vaisala HMP60 RH/T probes. This is stated in the original submission.

  - We note that accurate quantification of water vapour is only a central part of the actinometric calibration method but not the proxy method. The humidity of the CIMS exhaust is determined in order to characterize the humidity dependence of the instrument sensitivity, but the accuracy of this particular humidity measurement does not affect the measurement accuracy - all that matters is that the same measurement is used during calibrations as during ambient measurements.

  - Unfortunately we did not directly calibrate the RH probe used for the actinometric calibration. Shortly after the completion of the laboratory data presented in the manuscript we did compare the two RH/T probes to each other and found the $X_{H2O}$ values determined from the probes agreed within 3.5 %. This was accomplished by having both RH/T probes sample the same flow of zero air that was humidified over a wide range of values (0% to 80% RH). In addition, we have acquired a brand new Vaisala HMP 60 RH sensor which agrees to within 1.1 % of the sensor used to quantify HONO. The brand new sensors accuracy (manufacturer-stated accuracy = 3 %) combined with the results of these experiments gives us confidence that we can state the absolute accuracy as 5 %.

  - A supplement section focused on the parameters necessary to quantify HOx and HONO by chemical actinometry will be included in the revised manuscript. In this section the uncertainty that we ascribe to our $H_2O$ measurements (10 % $2\sigma$) is discussed in detail: We choose a [$H_2O$] uncertainty of 5 % ($2\sigma$) because the Vaisala HMP60 RH/T probe used for [$H_2O$] quantification in this manuscript had not recently been factory calibrated but agreed to within 1.1 % with a brand new Vaisala HMP60 sensor that has a manufacturer-stated accuracy of 3 %. In a separate experiment, the RH/T probe was compared to the CIMS exhaust RH/T probe (also a Vaisala HMP60) and agreed to within 3.5 %. The [$O_2$] uncertainty is based on the range stated by the gas manufacturer (Airgas), and the uncertainty of [$O_3$] is that of the CAPS NO2 instrument (see main text Sect. 2.2), which effectively measures Ox ($O_3 + NO_2$).

- State the make and model of the CAPS $NO_2$ instrument and scroll pump
  - The make and model of the scroll pump (Agilent Technologies IDP-7) was stated in the initial submission. The CAPS information was not specified. The CAPS is made by Aerodyne Research, Inc. and is simply referred to "CAPS $NO_2$ Monitor". Therefore, we will include the make in the revised manuscript (see below).

  - **Revised text:**
    A Cavity Attenuated Phase Shift (CAPS) spectrometer ( Aerodyne Research, Inc.) was used to detect $NO_2$ (Kebabian et al., 2008).

**Calibration methods**

- For reaction (R5), it is important that the fate of the H atom is only reaction with $O_2$. The H atoms be formed with excess energy via (R4) and Fuchs et al (2009) in a nice paper showed that 100% of the H atoms do result in HO2 formation rather than reacting via other potential exothermic reactions – this paper should be referenced. Fuchs et al., AMT, 2009, 2, 55.

- o Thank you for this suggestion. We have read through two papers by Fuchs et al. including the 2009 AMT paper listed in this comment. We have not found a discussion related to the fate of H atoms in either paper.

  - o A similar comment regarding the fate of the H atom is listed later in response to this reviewer. See page 7 for a discussion on the chemical kinetics of the H + O$_2$ reaction.

- Page 4, line 109. What is the length of the section of the flow tube exposed to the Hg lamp? State this – especially as the remaining length of the flow tube before sampling by the CIMS is stated later in the paper.
  - o We have updated the text to include the exposed length. The exposed portion is listed as an approximate because the tape covering the photolysis tube is not completely straight (i.e., exposed lengths range around the quartz cell from near 0.32 cm to near 0.62 cm).

  - o **Relevant text (updated portion is shown highlighted):**
    The humidified air-NO mixture is transported past the mercury lamp within a partially exposed 26.7 cm quartz tube (I.D. = 1.04 cm; approximately 0.5 cm exposed).

- Page 4, line 110 – state the range of lamp flux which was used to generate HONO (this can be estimated from the product of F x t divided by an approximate photolysis time).

  In the text we state: "HONO sample concentrations are controlled by adjusting the lamp flux with a Variac Variable transformer, adjusting the relative flow rates of the dry and humidified zero air/N$_2$, or adjusting the absolute flow rates to alter the lamp exposure time." We only varied lamp flux for the proxy calibration (specifically for a multipoint Fig. 3 calibration curve), and we note that for this calibration type it is not necessary to quantify Ft. We include these mentions of *Ft and F* in the actinometric calibration section Sect. 2.2.1:
  For typical operating conditions, the values of *F·t* and photon flux *F* (calculated using an approximate gas exposure time) are $3.48 \times 10^{12}$ photons cm$^{-2}$ and $7.1 \times 10^{14}$ photons cm$^{-2}$ s$^{-1}$, respectively.

**Actinometric calibration**

- Page 4, line 127, small w in where
  - o All instances of capital 'w' "Where" that appear after an equation have been changed to lowercase 'w' "where".

- Page 5, line 137 and following. For the use of the O2/O3 actinometry method for calibration of HOx instruments, where the gas mixture exiting the calibration flow tube is sampled at the centre of the flow by (usually) a pinhole and a supersonic expansion (e.g. for FAGE instruments), a profile factor (or P factor) needs to be used to reflect the laminar flow velocity profile across the flowtube (and hence a range of photolysis times) if the flow regime is laminar or partially turbulent. This is the case if the O3 measurement is taken from the remaining flow that does not enter the FAGE sampling pinhole which samples the OH. If the flow is fully turbulent, as often used for the N2O/NO actinometric method, then such a P-factor does not need to be taken into account as there is a flat velocity profile. In the case here, there is no pinhole for sampling, but there a short section of tubing before the excess flow goes to the CAPS instrument, and even if a profile factor does not need to be used, a discussion should be added

regarding the nature of the flow in the photolysis chamber/flowtube and whether this needs to be taken into account in any way in the calibration.

- o We have laminar flow (Reynolds number ≈ 600) up until the exit of the photolysis tube at which point the diameter is reduced from 1.1 cm (quartz tube) to 0.610 cm (3/8" OD PFA coupling tube). This 3/8" OD tube is then coupled to a PFA tee of smaller cross sectional area that splits the flow towards the CIMS and the CAPS. The PFA tee is arranged so that air travels straight to the CIMS and must make a 90° turn for the CAPS/vent line. The flows to the CIMS IMR (ion molecule reactor) and CAPS/vent line are 2.2 and 3 lpm, respectively. These sudden changes in the inner diameter and flow direction very likely introduce enough turbulence such that we can expect the CAPS and CIMS to be sampling well mixed air.

- o This has been addressed within the methods section (Sect. 2.2):
  The resulting calibration gas enters a PFA tee and is arranged so that the air travels straight to the CIMS (2.1 SLPM) while the remaining flow (~ 3.1 SLPM) makes a 90° turn for the CAPS line which includes a vent. The gas flow is initially laminar within the quartz photolysis tube (Reynolds number ≈ 600). This results in an initial [$HO_2$] (and therefore [HONO]) radial gradient in which the greatest concentrations exist near the flow tube walls (i.e., where the flow rates are lower and the UV exposure times longer). Turbulence is induced by the sudden changes in tube inner diameter at the quartz tube exit (reducing union) and upon entering the PFA tee. The air is therefore most likely well mixed prior to being split within the PFA tee. The excess flow within the CAPS line (~ 2 SLPM) was vented past an RH/T probe to determine the water mixing ratio in the photolysis cell.

- Page 5, line 140, state the value of the $O_2$ optical depth used in this calibration flow tube. Also, give some more further details about what is meant by "non-ideal overlap" between the lamp and the $O_2$ absorption spectrum.
  - o We have included additional information regarding the $O_2$ absorption spectrum/Hg lamp emission overlap (see text included below). We have also stated the $O_2$ concentration and column density within the photolysis tube center $O_2$ optical depth and $O_2$.

  - o Additions to text:
    The effective value for $\sigma_{O2}$ must be experimentally determined for the individual mercury lamp at the experimental $O_2$ optical depth. This is required because the 184.9 emission line, which comprises two peaks due to self-reversal, can vary from lamp to lamp and with operating conditions, and the $O_2$ absorption spectrum steeply decreases near the mercury lamp emission maximum (Lanzendorf et al., 1997). We use an experimentally determined $\sigma_{O2}$ value of $1.4 \times 10^{-20}$ $cm^2$ $molec^{-1}$ for the mercury lamp used for these experiments. The JPL-recommended value of $7.1 \times 10^{-20}$ $cm^2$ $molec^{-1}$ was used for $\sigma_{H2O}$ (Burkholder et al., 2020). The value of [$O_2$] is based on the flows mentioned in Sect. 2.2 and is equal to 20.1 ± 1.0 %, the $O_2$ optical depth is 0.033, and the $O_2$ column density (within the photolysis tube center) of $2.4 \times 10^{18}$ molecules $cm^{-2}$. For typical operating conditions, the value of $F \cdot t$ and an estimated photon flux $F$ (calculated using an approximate gas exposure time) are $3.48 \times 10^{12}$ photons $cm^{-2}$ and $7.1 \times 10^{14}$ photons $cm^{-2}$ $s^{-1}$, respectively. The value of [$O_3$] here was near 20 ppbv and determined with the CAPS $NO_2$ monitor after its reaction with NO, forming $NO_2$. This is measured with dry air flowing in the photolysis chamber so that the $NO_2$ measured is solely from the reaction of NO with $O_3$ and not $HO_2$. These $F \cdot t$ and [$O_3$] values are high compared to those used for most $O_3$ actinometry HOx calibrations, in which [$O_3$] is often less than 1ppbv (e.g., Faloona et al. (2004)), but are comparable to those used by Dusanter et al. (2008). High $F \cdot t$ values were used so that typical ambient HONO concentrations (ranging up to several ppb) could be prepared. [$H_2O$] is determined using the measured RH, temperature, and pressure. The uncertainties of the variables in Eq. (3) are discussed in Sect. S3 of the supplement. The combined uncertainty (2σ) for [HOx] (and therefore [HONO]) calculated using this equation is 29 % (see supplement for details).

- Page 5, line 146, in the equation, (1-(beta/(1+beta)) is the same as 1/(1+beta) and might be simpler to write?

    - Thank you for this suggestion. We have changed the equation and relevant text to match this.

- Page 5, line 165, when only a low concentration of $O_2$ is used, state both the concentration of $O_2$ and also the lifetime of the H atoms by reaction with this $O_2$ to show it is still very short.

    - As mentioned in a previous comment, ensuring proper conversion of H to $HO_2$ by reaction with $O_2$ is critical and we agree that the relevant kinetics should be discussed more quantitatively. We have used the JPL-recommended rate constant for the $H + O_2$ reaction to calculate pseudo-first order rate constants and H lifetimes ($\tau_H$) for the $[O_2]$ values used during both calibration types at $T = 298$ K and $M = 2.45 \times 10^{19}$ molec cm$^{-3}$.

        - Actinometric calibration conditions:
          $$[O_2] = 20.1\%, k_{pseudo} = 5.6 \times 10^6 \, s^{-1}, \tau_H = 1.8 \times 10^{-7} \, s$$

        - $NO_2$ proxy calibration conditions:
          $$[O_2] = 0.040\%, k_{pseudo} = 1.1 \times 10^4 \, s^{-1}, \tau_H = 8.9 \times 10^{-5} \, s$$

    - The calculated lifetimes, though three orders of magnitude apart, are both extremely short, and we safely conclude minimal interference by competing reactions (e.g., gas phase reactions with $NO_2$, surface reactions). In the revised text, we mention the $[O_2]$ values for both calibration types. We also list the pseudo-first order rate coefficient for the low $[O_2]$ case of the proxy calibration (shown below with revised text highlighted).

      The resulting low $O_2$ concentration ($[O_2] = 0.040 \pm 0.002 \, \%$) is sufficient for the full conversion of H to $HO_2$ (R5) but results in a negligible amount of $O_3$ formed by $O_2$ photolysis (R7-R8), confirmed by toggling the UV source on and off with dry N2 flowing. The pseudo-first order rate constant for the H to $HO_2$ conversion (R5) is $1.1 \times 10^4$ s$^{-1}$ for this $[O_2]$ value, ensuring that no other H atom reactions are competitive.

**Results and Discussion**

- Page 6, line 180. As well as the CIMS chi(H2O) which is the ratio of adduct to reagent, can the absolute water vapour concentration in the flow tube (as a mixing ratio) be stated also, as this will enable comparison with typically encountered levels of water vapour encountered in the atmosphere. Also, for completion, the total pressure of gas in the flow-tube and also the temperature should also be stated (it is on the figure caption)

    - Instances in which photolysis tube relative humidity values are stated have been updated to include mixing ratio, temperature, and pressure.

- Page 6, line 180, the slope is discussed to give the sensitivity, can some discussion also be made of the intercept to figure 3, which is presumably from HONO impurities in the N2 used?

    - The background values of HONO are from impurities within the NO flow rather than the $N_2$. All anonymous reviewers have made note of the HONO background signals and resulting Fig. 3 y intercept. We mention in the original text within the methods section that the NO addition leads to the enhanced background signal earlier in the manuscript. We have updated this text for clarity

(see below). We additionally include a statement regarding the Fig. 3 y-intercept in its caption. The updated caption is provided below.

- o **Updated Text of Sect. 2.2.1:**
  The CIMS response to HONO is determined by acquiring a background by briefly toggling off the mercury lamp. This background CIMS signal is humidity dependent, so a background is taken at each humidity setting. The high background CIMS $I(HONO)^-$ signals are mainly from impurities in the NO flow.

- o **Updated Fig. 3 caption:**
  Figure 1: Calibration curve obtained using the $NO_2$ proxy calibration method at a constant humidity (RH = 17.7%, $[H_2O]$ = 0.388% as measured in the CIMS scroll pump exhaust) by varying photon flux F with a Variac variable transformer. Error bars represent ± 2σ uncertainty. The data was fitted using the York bivariate regression method (York et al., 2004). The y-intercept (3,820 ncps) represents the background CIMS $I(HONO)^-$ signal during this calibration and is mostly from impurities in the constant NO addition. Without the NO addition (for ambient sampling) the typical background signal is 75 ncps.

- Page 6, line 199. I agree that the chi(H2O) is a useful quantity to compare CIMS instruments, but the level of water vapour (as a mixing ratio) also needs to be stated in order to gauge how the sensitivity of the instrument varies for different regions of the atmosphere.
  - o We have included the mole fraction within the IMR because this is the parameter that we would use to track the sensitivity. In the revision we will include the range of photolysis tube $H_2O$ mole fractions (and relative humidities) to provide the desired information that relates ambient conditions (RH, T, etc.) to our specific CIMS IMR $[H_2O]$ values. We note that the relationships among ambient $[H_2O]$, CIMS IMR $[H_2O]$, and CIMS sensitivity are not universal for all CIMS instruments.

- Page 7, Figure 2. Comment on the slower fall and rise of the $NO_2$ signal compared with that of HONO when the lamp is toggled off.
  - o The Fig. 2 time series has been changed to show results acquired during experiment performed on a different day. This is mentioned on page 1 of this document and explained in response to reviewer #3 (See page 17 of this document). The mentioned slow response in $NO_2$ signal still occurs in the updated figure and is simply due to the CAPS $NO_2$ instrument having a slower instrumental time response than the CIMS. The CAPS time response is determined by its volumetric flow rate and the volume of the absorption cell. This and other details regarding the CAPS instrument can be found in the reference given (Kebabian et al. 2008).

- Figure 2, also, state T and P conditions. State also the RH and absolute mixing ratio of $H_2O$ for this experiment.
  - o This has been included within the revised text. Figure 2 (the time series figure) has been updated as noted on page 1.

- Page 8, Figure 3, the RH and $[H_2O]$ are measured in the CIMS scroll pump exhaust. Was $H_2O$ also measured closer to the exit of the photolysis region to check that the RH did not change (e.g. as a result of any temperature change after the pump or wall-losses of $H_2O$?) The measurement of water vapour is critical to the calibration.
  - o Measuring the RH and $[H_2O]$ within the scroll pump exhaust is specifically for gauging the CIMS response. We choose this location because $[H_2O]$ is different in the CIMS ion molecule reactor

(IMR) than in the photolysis tube due to dilution by the dry reagent flow and addition of the humidified $N_2$. A second RH/T probe was used for the measurement of water vapor within the photolysis tube (without any dilution).

- Figure 3, discuss the intercept in the text.
    - A response to a similar comment was included in this section (see response to point 2 under *'Results and Discussion'* comments)

- Figure 3, the dotted line is clearly a linear least squares fit to the data, but this needs to be added to the caption.
    - We have fitted the line using the York bivariate method. This was stated in the text. The Fig. 3 caption, included below has been updated to include this information (new text shown highlighted):

        **Figure 2**: Calibration curve obtained using the $NO_2$ proxy calibration method at a constant humidity (RH = 17.7%, [$H_2O$] = 0.388% as measured in the CIMS scroll pump exhaust). Error bars represent $\pm 2\sigma$. The data was fitted using the York bivariate regression method (York et al., 2004). The y-intercept (3,820 ncps) represents the background CIMS I(HONO)$^-$ signal and is mostly from impurities in the constant NO addition.

- Figure 3. From the slope the sensitivity factor is obtained. Can the limit of detection of the instrument also be stated from the calibration and associated noise levels? Was a multipoint calibration also performed using the actinometric method?
    - The multipoint calibration was performed for the $NO_2$ proxy calibration only.

    - The limit of detection for outdoor measurements can be determined using the slope of this curve along with the true background signal observed during sampling (i.e., without the NO addition). The calibration technique, however, causes an offset in the CIMS I-HONO$^-$ signal due to impurities introduced by the NO (i.e., the y intercept is not representative of the true background). During sampling near Boise, Idaho, background I-HONO$^-$ counts were ~76 normalized counts per second (ncps). This led to a detection limit of 8.0 pptv for HONO at SNR = 2 (time averaging of 1 s). This information on the detection limit has been included in the supplement.

        The signal-to-noise ratio (SNR) and limit of detection (LOD) for our CIMS measurements of HONO are determined by the precision of the total and background signals, both of which are dominated by shot noise. The SNR is given by SNR = $(S_T - S_B)/(\sigma^2_T + \sigma^2_B)^{1/2}$ where $S_T$ is the total signal, $S_B$ is the background signal, $\sigma_T$ is the precision of the total signal, and $\sigma_B$ of the background signal. For our lab experiments the background signal was elevated due to HONO impurities associated with the high NO mixing ratios used (1.58 ppmv). For field measurements from 2019 in Boise, Idaho, our unnormalized I(HONO)$^-$ background signal was 228 counts/s, corresponding 76 ncps. With a sensitivity of 2 ncps ppt$^{-1}$ (that of a typical ambient humidity), the resulting 1 s LOD was 8.0 pptv (SNR = 2). An upcoming manuscript focused on the 2019 field measurements will have further discussion of limit of detection.

- Page 9, figure 4. Discuss the shape of the graph and the possible reasons for this shape and increasing sensitivity at lower chi($H_2O$)
    - The general trend in sensitivity with [$H_2O$] is mentioned in the manuscript: "Sensitivities ranged from 1.5 to 5.3 ncps ppt$^{-1}$ with the greatest values observed at low $\chi_{H2O}$ settings".

o A discussion of humidity effects on ionization chemistry will be included in a new supplement section titled *"Chemical Ionization Mass Spectrometry Humidity Effects"*. The anticipated text in that section is included below. The supplement section will be referred to twice: 1.) during the methods section focused on instrumentation and 2.) the results and discussion section focused on Fig. 4 (the sensitivity vs CIMS IMR [$H_2O$] plot).

The ionization chemistry utilized is sensitive to humidity. HONO can be ionized by reaction with either $I^-$ or the iodide-water adduct $I(H_2O)^-$ to form the detected adduct $I(HONO)^-$:

$$I^- + HONO \rightarrow I(HONO)^- \hspace{4cm} \text{RS1}$$

$$I(H_2O)^- + HONO \rightarrow I(HONO)^- + H_2O \hspace{2.5cm} \text{RS2}$$

The reverse reaction of S2 also occurs:

$$I(HONO)^- + H_2O \rightarrow I(H_2O)^- + HONO \hspace{2.5cm} \text{RS3}$$

Reactions RS2 and RS3 lead to variation in the sensitivity depending on ambient water vapor levels. The CIMS sensitivity to HONO decreases with ambient water vapor concentration as shown in Fig. 4 of the main text. Many compounds sampled by $I^-$ CIMS exhibit similar sensitivity-water vapor trends, though to varying degrees (Lee et al., 2014). As a result, sampling in dry conditions with low ambient water vapor mixing ratios allows for more sensitive detection of many compounds by $I^-$ CIMS including HONO. Drastic changes in sensitivity from atmospheric variability can be suppressed by constant dilution of the IMR with humidified nitrogen (as performed here) or by maintaining constant water vapor concentration (Veres et al., 2020).

- As well as IMR chi($H_2O$), the x-axis also should have the mixing ratio of $H_2O$ vapour in the photolysis tube used to provide HONO to the instrument. This will allow how the instrument sensitivity changes with ambient water vapour levels.
    - o Rather than assigning $X_{H2O}$ within the photolysis tube for all presented CIMS IMR $X_{H2O}$ values, we will include the range in photolysis tube $X_{H2O}$ for the Fig. 4 results (text to be included in the revision is shown below). We will also state the range in RH, the average temperature, and the pressure. These values allow readers to get an idea of how sensitivity corresponds to ambient $X_{H2O}$ or RH at our T/P conditions.
    The CIMS IMR $\chi_{H2O}$ values ranged from $1.77 \times 10^{-3}$ to $8.25 \times 10^{-3}$ and corresponded to a photolysis cell RH range of 4.1 to 71 % and photolysis cell $\chi_{H2O}$ values of $0.93 \times 10^{-3}$ to $16 \times 10^{-3}$ (average T = 19.6 °C; P = 760 Torr).

- Does the shape of the curve have any implications for use of this type of instrument in various regions of the atmosphere? Although not the primary focus of the paper, which is about the agreement of the 2 methods of calibration, which is very good, the shape of the calibration plot with water vapour will be of interest, and might have implications for measurement of HONO using CIMS.
    - o The shape of the curve indicates that the CIMS is most sensitive to HONO in dry conditions as stated in the original submission: "Sensitivities ranged from 1.5 to 5.3 ncps ppt[-1] with the greatest values observed at low $\chi_{H2O}$ settings". The sensitivity of most compounds measured by $I^-$ CIMS is humidity-dependent so tracking the humidity or attempting to maintain constant humidity in the IMR is common among CIMS users and has been described in papers that have focused on the technique itself (e.g., Lee, Lee, Veres). We have added content on the ionization chemistry responsible for humidity dependences as described earlier in our response. We refer to this section twice: 1.) during the methods section focused on $I^-$ CIMS and 2.) during the discussion of the sensitivity-humidity plot (Fig. 3).

- The error bars for the proxy method multipoint calibration (dark blue point) is similar to the error bar for the adjacent single point calibration method (light blue point), whereas I might have expected it to have been smaller given it is based on a slope of several points. Is there any reason for that?
  - The error bars for the multipoint calibration point (dark blue) are actually slightly smaller than the similar single point proxy calibration sensitivity results. This would be more easily observed if the multipoint calibration had been done at a [$H_2O$] value equal to that of one of the single point calibration points.

**Supplement**

- Line 9, small w in "where"
  - This has been updated accordingly.

- Variables in the text, for example *T* and *P* should be in italics (as they are in the equations)
  - This has been updated accordingly.

- The supplement is quite short, and consideration might be given to combining this with the main paper (which is fairly short).
  - Respectfully, we prefer to keep a supplement as we have strived to optimize the readability of the main text. The details regarding the derivation of the calculations and error propagation are cumbersome for the main text.

**Anonymous Reviewer # 2:**

General comments

- The manuscript would benefit from a critical discussion of the new method. For example, how does this new calibration method compare to existing ones?

  o Within the introduction, we discuss the Febo (acid displacement) method (in paragraph 4) and the photolytic methods (in paragraph 5). The setup and requirements for each calibration along with their corresponding warm-up times, possible quantification methods, and general calibration uncertainties. The concentration range prepared by the Febo method is also listed, and we mention the range in prepared [HONO] values by photolytic methods in the results and mention the range within the abstract. More information regarding both the Febo and photolyic methods has been included in this section to allow for a better comparison. The paragraphs as presented in the revised submission are provided below, and additions relevant to this comment are highlighted.

  Calibrations for HONO are challenging as this compound is not commercially available and rather must be prepared in situ. Most commonly, HONO is prepared by reacting hydrogen chloride vapor with sodium nitrite (Febo et al., 1995):

  $$HCl_{(g)} + NaNO_{2(s)} \rightarrow HONO_{(g)} + NaCl_{(s)} \tag{R3}$$

  This method presents several challenges. A stable source of HCl is required, usually from a heated aqueous solution, a gas cylinder, or a permeation tube. Consistent mixing between the HCl and the $NaNO_2$ powder is required. These calibrations also require substantial warmup times (often hours) to ensure source stability, though some recent versions report faster warmup periods (e.g., < 10 min reported by Villena and Kleffmann (2022)). High HONO concentrations (above 1 ppmv) are often produced, requiring dilution, though the temporary unrealistic HONO concentrations can lead to significant HONO loss by its self-reaction and inaccurate HONO quantification. A recent, noteworthy version of this calibration improves upon this concentration issue and has the ability to produce [HONO] on the order of tens of pptv (Lao et al., 2020). The generated HONO can be quantified by various methods including theoretical calculation (Villena and Kleffmann, 2022), conversion to NO followed by chemiluminescence detection (Lee et al., 2012; Lao et al., 2020; Villena and Kleffmann, 2022), thermal conversion to $NO_2$ followed by $NO_2$ quantification (Gingerysty and Osthoff, 2020), and conversion to aqueous nitrite followed by derivatization and detection by UV-vis (Peng et al., 2020). The calibration uncertainty depends on the output stability of the HONO source and the quantification technique used. Villena and Kleffmann (2022) demonstrate using two separate techniques that overall calibration uncertainties can be well below 10 % (2σ).

  More recently, photolytic HONO sources have been utilized. Humidified air is exposed to ultraviolet (UV) light to photolyze $H_2O$ to produce an equal mixture of OH and $HO_2$, which in the presence of excess NO then converts to HONO. This HONO output is stable within seconds (i.e., the initial UV lamp warm up time) and is tunable by altering humidity, UV flux, or UV exposure time. The HONO formed has been quantified based on the water vapor mixing ratio, water vapor absorption cross section, the UV flux, and the UV exposure time. This quantification approach thus far has been used to calibrate HONO photo-fragmentation instruments that detect OH using laser-induced fluorescence spectroscopy (Dyson et al., 2021; Bottorff et al., 2021). The HONO formed from a photolytic source has also been quantified by thermal dissociation followed by measurement of the $NO_2$ produced (Veres et al., 2015). These methods have an uncertainty of 30 to 36% (2σ), similar to the uncertainty for HOx calibrations based on water vapor photolysis (Dusanter et al., 2008). In this manuscript, we present an alternative photolytic HONO calibration that we refer to as the "$NO_2$ proxy" method. This method requires a direct $NO_2$ measurement that

is used as a 'proxy' to quantify HONO concentrations. We compare this new proxy calibration to the more standard photolytic calibration method as performed by Bottorff et al. (2021) and Dyson et al. (2021). This method has a lower uncertainty (typically ~10%, 2σ) and unlike the actinometric method does not require characterization of the mercury lamp emission spectrum.

- The manuscript would also benefit from sample data on how the new calibration method performs in the field. How stable is the calibration source (e.g., how often does the Hg lamp need to be recalibrated)?
    - We do not have field data to present here. We have measured ambient HONO in the field using the actinometric method only. That data will be included in an upcoming paper focused on HONO photochemistry that is to be submitted in the next few months. While we have not tested the long-term stability of the $NO_2$ proxy method yet, we expect that both methods will perform fine in the field as they are very similar to the traditional HOx calibration method that has been used for decades (as we have stated within the abstract, intro, and conclusion sections). Regarding lamp stability and need for Hg lamp recalibration (i.e., the value of effective absorption cross section of $O_2$), one major benefit of our novel proxy calibration method, which we note several times in the paper, is that it is not affected by long-term variations in UV lamp flux and there is no need to quantify the lamp-specific effective $O_2$ absorption cross section.

Specific comments
- Figure 2: The HONO background signal is very large. Why is that?
    - The appearance of background HONO was stated in the Sect. 2.2.1 of the original text and has been reworded to "Background CIMS I(HONO)- signals are elevated during calibrations due to impurities in the NO flow". However, all reviewers have noticed and questioned the background signal. In response we have added an additional sentence and for caption of the Fig. 3 (multipoint calibration curve). The edited caption is provided here and new text is highlighted.

    - **Fig. 3 caption:**
    Figure 3: Calibration curve obtained using the $NO_2$ proxy calibration method at a constant humidity (RH = 17.7%, [$H_2O$] = 0.388% as measured in the CIMS scroll pump exhaust). Error bars represent ± 2 σ. The data was fitted using the York bivariate regression method (York et al., 2004). The y-intercept (3,820 ncps) represents the background CIMS I(HONO)$^-$ signal during this calibration and is mostly from impurities in the constant NO addition. Without the NO addition (for ambient sampling) the typical background signal is 75 ncps.

- Figure 4: Why is the precision so low at low water concentration?
    - There are larger error bars for the lower water vapor concentrations of the $NO_2$ proxy calibration shown for the Fig. 4 CIMS sensitivity plot. This occurred because smaller HONO concentrations were prepared with smaller [$H_2O$] (i.e., less water vapor was photolyzed). Smaller [HONO] values led to much smaller amounts of $NO_2$ formed, which made the relative error associated with its detection by CAPS increase substantially. This is explained in the original text (see below), and we also state that using higher [HONO] values (by increasing the photon flux with a Variac or exposing a larger section of the photolysis cell to the 185 nm Hg lamp emission) at low humidity would decrease these error bars.

    - **Original Text For Reference:**
    The uncertainty at the lowest $\chi_{H2O}$ settings, which were also the lowest HONO concentrations, were dominated by [HONO] quantification stemming from the background subtraction of $NO_2$. Therefore, the uncertainty can be minimized by using higher HONO concentrations. The presented 2σ uncertainty in using the proxy method ranged from 8.7 to 38.8%. All presented

proxy calibration uncertainties besides the two at lowest $\chi_{H2O}$ values (i.e., the smallest [HONO] values) fall well below the 30% 2σ uncertainty associated with the standard $O_3$ actinometry calibration. The proxy calibration is therefore a general improvement over the standard actinometry calibration, especially at high [HONO] values.

**Anonymous Reviewer #3:**

**Comments:**

- The authors mention the use of two photolytic HONO calibration methods. It seems to this reviewer that only one calibration method is used in this publication, namely the water photolysis method for the generation of OH and HO2 radicals with the implementation of an additional step to convert both OH and HO2 into HONO. There are however two different approaches used to quantify the amount of generated HONO. The authors should revise the text accordingly.

  > We consider a calibration method to involve two components. The first component is the technique for generating or supplying an analyte to a detector. The second component is the quantification method. The two methods presented are slightly different in the method of HONO generation (i.e., the carrier gas used – air vs. $N_2$ with trace $O_2$) and the quantification methods are very different. We believe it is appropriate to refer to them as two separate calibration methods, though as noted in the text they have many similarities.

- L41-44 : " For a Beijing, China based study, a comparison of several HONO measurements showed an overall mixed agreement with major differences observed for a few techniques (Crilley et al., 2019). Measurements in Houston, Texas showed several instruments to mostly agree in capturing variations in HONO, though there were differences in the magnitude of presented [HONO] values (Pinto et al., 2014)." – Please indicate the level of disagreement.

  - We have changed this text and added additional information regarding the levels of disagreement: A comparison of several HONO measurements in Beijing showed an overall mixed agreement with a few instruments disagreeing by more than a factor of two (Crilley et al., 2019). Measurements in Houston, Texas showed overall good agreement (within 20 %) between most instruments with larger differences of over 100 % observed for one of the instruments for some time periods (Pinto et al., 2014).

- L74-76: "The CAPS instrument was calibrated using a 2B 75 Technologies Model 306 O3 Calibration Source, which agreed to within 2.5% with a Thermo Environmental Instruments 49C O3 Calibrator. We assign an uncertainty of 3% (2σ) to the NO2 measurements" – The authors indicate that the CAPS instrument is calibrated using an O3 calibration source. How was the gas phase titration of O3 into NO2 performed? The stated 2σ uncertainty of 3% is rather low. How was it inferred?

  - We have added more text to clarify the $NO_2$ calibration technique and the $NO_2$ measurement uncertainty. We agree that the stated uncertainty was likely not appropriate and have increased the stated 2σ uncertainty to 4 %.

  - **Relevant updated text:**
    A Cavity Attenuated Phase Shift (CAPS) spectrometer (Aerodyne Research, Inc.) was used to detect $NO_2$ (Kebabian et al., 2008). The CAPS also indirectly measured $O_3$ as it was converted to $NO_2$ by reaction with excess NO. The CAPS instrument was calibrated using a 2B Technologies Model 306 $O_3$ Calibration Source. Ozone outputs were varied between 10 and 300 ppbv with greater than 99.99 % conversion efficiency to $NO_2$ by reaction with excess NO ([NO] = 1.82 ppmv) within approximately 15 m of FEP tubing (i.d. = 0.476 cm; residence time = 17.1 s, pseudo-first order rate constant of 0.8 s$^{-1}$). The manufacturer stated accuracy of this $O_3$ calibrator is 2% (2σ), though no recent factory calibrations have been conducted. Therefore, a second calibration was conducted with a Thermo Environmental Instruments 49C $O_3$ Calibrator, which agreed to within 2.5%. We assign an uncertainty of 4 % (2σ) to the $NO_2$ measurements to account for possible drift in accuracy.

- L88-90: "We account for the humidity dependence of the instrumental response by determining the mole fraction of H2O(g) (χH2O) in the IMR by measuring the RH and temperature of the IMR in the exhaust of the scroll pump." – Why is there a humidity dependence?
  - We have added a new section to the supplement titled *Chemical Ionization Mass Spectrometry Humidity Effects* (see quoted texter on page 10 of this response document). We refer to this section twice: once within the methods section and once discussing the sensitivity vs humidity figure (Fig. 4).

- L140: "We use a value of $1.4 \times 10^{-20}$ cm$^2$ molec$^{-1}$ for $\sigma O_2$ for the mercury lamp used for these experiments." – Was $\sigma O_2$ determined experimentally? If not, how did the authors estimate the uncertainty associated to $\sigma O_2$? Was it factored in error bars shown in Fig. 4 for the actinometric approach?
  - The text has been updated accordingly to make clear that $\sigma_{O2}$ was experimentally determined (see text below), and additional information is provided about the lamp emission profiles as requested by reviewer #1.

  - **Relevant text  (new text shown highlighted):**
    The effective value for $\sigma_{O2}$ must be experimentally determined for the individual mercury lamp at the experimental $O_2$ optical depth. This is required because the emission profile near 14.9 nm, which comprises two peaks due to self-reversal, can vary from lamp to lamp and with operating conditions, and the $O_2$ absorption spectrum steeply decreases near the mercury lamp emission maximum (Lanzendorf et al., 1997). We use an experimentally determined $\sigma_{O2}$ value of $1.4 \times 10^{-20}$ cm$^2$ molec$^{-1}$ for the mercury lamp used for these experiments. The JPL-recommended value of $7.1 \times 10^{-20}$ cm$^2$ molec$^{-1}$ was used for $\sigma_{H2O}$ (Burkholder et al., 2020).

  - The error bars for the Fig.4 actinometric calibration results have been updated. They error bars in CIMS sensitivity combine the $\pm 29\%$ ($2\sigma$) uncertainty for the [HONO] quantification with the 15 s CIMS precision. Previously, the error bars included only the uncertainty for the [HONO] quantification. New text and a table have been added to the SI to detail the individual uncertainties of chemical actinometry variables that combine to the 29 % ($2\sigma$) for quantified [HONO] values.

  - **New Supplement Section:**
    **S3 Uncertainty Propagation for Actinometric Calibration**

    The uncertainty in [HONO] quantified by the actinometric method involves adding in quadrature the relative uncertainties of each variable in Eq. (3) (see Sect. 2.2.1 of main text) as shwon in Table S1. The resulting [HONO] relative uncertainty is 26.9 % ($2\sigma$). A small uncertainty is also associated with the term $\beta$ used in the correction equation that converts [HOx] to [HONO] (Eq. (4) in the main text). Because of this small correction, we round up and assign an overall uncertainty of 27 % ($2\sigma$) to the calibration. The $\sigma_{O2}$ uncertainty is the largest contributor to this overall [HONO] uncertainty and was determined experimentally. The uncertainty in $\sigma_{H2O}$ was the same as used by Dusanter et al. (2008). We choose a [H$_2$O] uncertainty of 5 % ($2\sigma$) because the Vaisala HMP60 RH/T probe used for [H$_2$O] quantification in this manuscript had not recently been factory calibrated but agreed to within 1.1 % with a brand new Vaisala HMP60 sensor that has a manufacturer-stated accuracy of 3 %. In a separate experiment, the RH/T probe was compared to the CIMS exhaust RH/T probe (also a Vaisala HMP60) and agreed to within 3.5 %. The [O$_2$] uncertainty is based on the range stated by the gas manufacturer (Airgas), and the

uncertainty of [O$_3$] is that of the CAPS NO2 instrument (see main text Sect. 2.2), which effectively measures Ox (O$_3$ + NO$_2$).

- **Table S1 Relative Uncertainties (2σ) for Variables Used to Quantify [HONO] By Chemical Actinometry**

| Variable | Relative Uncertainty (%) |
|---|---|
| [H$_2$O] | 5 |
| σ$_{H2O}$ | 6 |
| [O$_3$] | 4 |
| [O$_2$] | 5 |
| σ$_{O2}$ | 25 |
| [HONO] | 26.96 |

- Fig. 2: Please indicate the HONO mixing ratio derived from NO2 in the caption.
  - There were issues with this original time series figure (see the response to the next comment). Briefly the optimized [NO] that we used to obtain the results for all other figures was not used here (i.e., the time series data is of an early experiment). Figure 2 has since been remade (shown in response to the next comment), and we indicate the new HONO mixing ratio within the caption.

- Fig. 3: Why is there a significant y-intercept when HONO=0 in Fig. 3? This intercept is approx. 3000 ncps, while in Fig. 2, measurements performed without HONO provide a normalized background signal of approx. 1000 ncps. Why are these "background" signals different? From Fig. 3, the normalized CIMS signal extrapolated for a HONO mixing ratio of approx. 5000 ppt (similar to that generated in Fig. 2, ΔNO2 of approx. 2500 ppt à HONO of approx. 5000 ppt) would be approx. 17500 ncps, which is approx. 14500 ncps after subtraction of the "background" signal. This does not compare to that reported in Fig. 2 since the background subtracted CIMS signal is approx. 5000 ncps. This difference of a factor of 2.9 does not seem to be only due to the humidity-dependence of the CIMS response reported in Fig. 4 since the sensitivity decreases by a factor of 1.4 when humidity varies from 0.39% to 0.56% (water mixing ratio estimated by the reviewer for data shown in Fig. 2, estimation based on reported RH values for the photolysis cell and assuming that temperature was the same for experiments displayed in Figs. 2 and 3). Could the authors comment on this?

  - This Fig. 3 y-intercept has been questioned by the other reviewers. The Fig. 3 y-intercept occurs due to background [HONO] exists from impurities in the NO. While this is stated in the text of the original submission within the methods section, we have restated for clarification. We have also included a new sentence in the caption of the Fig. 3 multipoint calibration curve where we state the value of intercept (i.e., general background signal) and its source being the impurities in our NO flow. This is explained in response to Reviewer 1, see page 8 of this document.

  - Your observation regarding the especially low CIMS background HONO signal in Fig. 2 is correct. The original version of Figure 2 used data collected in an early experiment that used too low of a [NO] value (nearly a factor of 4 lower than we state in the paper). The smaller [NO]

concentration led to reduced sensitivities because of OH losses (e.g., to collisions with tubing) that were much greater than $HO_2$ losses.

- We have updated figure 2 using data collected with the final optimized [NO] value. The new data section shows a slightly smaller [HONO] value (approximately 3,000 pptv) and a similarly timed mercury lamp toggle. We obtained this section during a single-point calibration as performed for Fig. 4. A different relative humidity within the photolysis cell of 29% was used. Mentions of the previous humidity have been updated. The background counts here are also close to the y-intercept of equation 3 – near 4,000 ncps - though they do not perfectly match due to the different humidity.

**Updated Fig. 2 and Caption:**

[Figure]

**Figure 2: One-second averaged time series data for a proxy calibration at a constant relative humidity. The iodide HONO adduct signal (cps) is shown normalized per one million reagent ions (ncps). The highlighted section represents the period in which the 184.9 nm mercury lamp is toggled off to obtain background [NO₂] and HONO signal. The NO₂ is shown with an offset so that background values are near 0 ppbv. The resulting difference in NO₂ indicates that approximately 3,000 pptv [HONO] is sampled by the CIMS.**

**Edits:**

- L111: "humidified ZA/N$_2$" should read "humidified zero air/N2"
  - The text has been changed accordingly.

- L136: "ΦO$_2$" should read "ΦO$_3$"
  - This text has been changed accordingly. Thank you for pointing this typo.

- L171: "40% within the H$_2$O photolysis cell" & L178 "28% within the photolysis cell and 18% within the CIMS IMR" – Please indicate the temperature for each RH measurement. Other instances in the text. Temperature should be provided each time RH is reported.
  - This was similarly requested by Reviewer #1, and temperatures have since been provided.

- Fig. 2 vs. Fig.3: please use the same title for the y-axis
  - The Fig. 2 (time series) y-axis is titled "HONO signal (ncps)" while the Fig. 3 (multipoint calibration) y axis is titled "CIMS signal (ncps)". We agree with keeping these consistent to avoid any possible confusion to readers. They are now consistent and read as "I(HONO)$^-$ signal".

**Supplementary material:**

- L5-6: "(kR7b/kR7a" should read "(kR6b/kR6a" & "(kR7b/(kR7a + kR7b)" should read "(kR6b/(kR6a + kR6b)"
  - Thank you for noticing this typo. The mentioned supplementary text has been updated. This issue has also been corrected for in supplementary equations S4 and S6.